# O-GlcNAcylation of nuclear proteins in the mouse liver exhibit daily oscillations that are influenced by meal timing

Xianhui Liu[1,2,3☉], Yao D. Cai[2☉], Chunyan Hou[4], Xu Liu[1], Youcheng Luo[1], Aron Judd P. Mendiola[5], Xuehan Xu[2], Yige Luo[6], Haiyan Zheng[7], Caifeng Zhao[7], Ching-Hsuan Chen[2], Yong Zhang[1], Yang K. Xiang[3], Junfeng Ma[4], Joanna C. Chiu[2*]

1 Cambridge-Suda Genomic Resource Center, The Fourth Affiliated Hospital, Suzhou Medical College, Soochow University, Suzhou, China, 2 Department of Entomology and Nematology, University of California, Davis, California, United States of America, 3 Department of Pharmacology, University of California, Davis, California, United States of America, 4 Department of Oncology, Lombardi Comprehensive Cancer Center, Georgetown University Medical Center, Washington, DC, United States of America, 5 Department of Medical Microbiology and Immunology, University of California, Davis, California, United States of America, 6 Department of Evolution and Ecology, University of California, Davis, California, United States of America, 7 Biological Mass Spectrometry Facility, Robert Wood Johnson Medical School and Rutgers, the State University of New Jersey, Piscataway, New Jersey, United States of America

☉ These authors contributed equally to this work.
* jcchiu@ucdavis.edu

## Abstract

The liver circadian clock and hepatic transcriptome are highly responsive to metabolic signals generated from feeding–fasting rhythm. Previous studies have identified a number of nutrient-sensitive signaling pathways that could interpret metabolic input to regulate rhythmic hepatic biology. Here, we investigated the role of O-GlcNAcylation, a nutrient-sensitive post-translational modification (PTM) in mediating metabolic regulation of rhythmic biology in the liver. We observe daily oscillation of global nuclear protein O-GlcNAcylation in the liver of mice subjected to night-restricted feeding (NRF) using label-free global O-GlcNAc proteomics. Additional site-specific O-GlcNAc analysis by tandem mass tag mass spectrometry further supports temporal differences in O-GlcNAcylation by revealing day–night differences. Proteins involved in gene expression are enriched among rhythmically O-GlcNAcylated proteins, suggesting rhythmic O-GlcNAcylation may directly regulate the hepatic transcriptome. We show that rhythmic O-GlcNAcylation can also indirectly modulate nuclear proteins by interacting with phosphorylation. Several proteins harboring O-GlcNAcylation-phosphorylation interplay motif exhibit rhythmic O-GlcNAcylation and phosphorylation. Specifically, we show that O-GlcNAcylation occurs at a phospho-degron of a key circadian transcriptional activator, circadian locomotor output cycles kaput (CLOCK), thus regulating its stability and transcriptional output. Finally, we report that day-restricted feeding (DRF) in the nocturnal

**Data availability statement:** All data needed to evaluate the conclusions in the paper are present in the paper and/or the Supporting information. All individual numerical values that underlie the data summarized in Figs 1, 5, 6, and S7 are available in the S1 Data file. The raw metabolomics dataset has been deposited in the open metabolomics database, Metabolomics Workbench (https://www.metabolomicsworkbench.org/), under accession no. [ST002952] and [ST002950]. The raw label-free global O-GlcNAc proteomics, TMT-phosphoproteomics and proteomics datasets have been deposited into MassIVE repository (MSV000094417) (MassIVE: ftp://massive.ucsd.edu/v07/MSV000074417). Data for TMT site-specific O-GlcNAc proteomics (MSV00097856) (MassIVE: ftp://massive.ucsd.edu/v09/MSV000097856) were also deposited and made public.

**Funding:** This work is supported by U.S. National Institutes of Health grants R01DK124068, R56DK124068, and R21AG082480 to J.C.C. and Ministry of Science and Technology, PRC, Science and Technology Innovation 2030 (STI2030)-Major Projects 2021ZD0203400 to YZ. The funders had no role in study design, data collection and analysis, decision to publish, or preparation of the manuscript.

**Competing interests:** The authors have declared that no competing interests exist.

**Abbreviations:** AMPK, adenosine-monophosphate-activated protein kinase; BMAL1, brain and muscle Arnt-like protein-1; CLOCK, circadian locomotor output cycles kaput; DRF, day-restricted feeding; FXR, farnesoid X receptor; HBP, hexosamine biosynthetic pathway; NRF, night-restricted feeding; PTM, post-translational modification; RAIN, rhythmic analysis incorporating nonparametric; SCN, suprachiasmatic nuclei.

mouse significantly alters O-GlcNAcylation pattern. Whereas global O-GlcNAcylation analysis indicates dampening of global O-GlcNAcylation rhythm in mice fed under DRF, site-specific analysis reveals differential responses of O-GlcNAc sites when timing of food intake is altered. Notably, a substantial number of O-GlcNAcylation sites exhibit inverted day–night profiles when mice are subjected to DRF. This suggests the dysregulation of daily nuclear protein O-GlcNAcylation rhythm may contribute to the disruption in liver transcriptome previously observed in DRF condition. In summary, our results provide new mechanistic insights into metabolic regulation of hepatic transcriptional regulators via interplay between O-GlcNAcylation and phosphorylation and shed light on the deleterious effects of improper mealtimes.

## Introduction

Circadian clocks are cell-autonomous, endogenous molecular timers that integrate environmental time cues and metabolic signals, enabling organisms to adapt to 24-h environmental changes by exhibiting time-of-day specific physiology and behavior, such as sleep–wake cycles, feeding–fasting cycles, and metabolism [1,2]. Clock disruption is associated with a wide range of diseases, including cancer, diabetes, and heart disease [3–6]. At the molecular level, mammalian circadian clocks are composed of interlocked transcriptional–translational feedback loops. At the core loop, brain and muscle Arnt-like protein-1 (BMAL1) and circadian locomotor output cycles kaput (CLOCK) form heterodimers and activate the expression of a repertoire of genes [7]. Among these clock-regulated genes are core clock genes that encode PERIOD (PER1, PER2 and PER3) and CRYPTOCHROME (CRY1 and CRY2) proteins, which form repressor complexes to inhibit the activity of BMAL1 and CLOCK [7]. As PER and CRY proteins degrade, transcriptional repression is relieved to initiate the next circadian transcriptional cycle. This cell-autonomous molecular clock was first characterized in the suprachiasmatic nuclei (SCN) of the hypothalamus in mammalian brain; the SCN receive light/dark signals to regulate the pace of molecular clocks [8–10]. Follow-up investigations have established that almost all peripheral tissues harbor molecular clocks, and metabolic signals from daily feeding activity appear to be more potent than light as time cues in peripheral tissues [10–13].

Among all the peripheral organs that are synchronized by feeding–fasting cycles, the liver is one of the most nutrient-sensitive [11,13–15]. Feeding nocturnal mice during daytime can reverse the phase of clock gene expression in liver within a few days [14,15]. Indeed, many studies revealed the importance of metabolic regulation of the molecular clock [16]. Adenosine-monophosphate-activated protein kinase (AMPK), a nutrient sensor, phosphorylates and destabilizes CRY1 [17]. $NAD^+$ is essential for the activity of Poly [ADP-ribose] polymerase 1 (PARP1), which poly(ADP-ribosyl)ates CLOCK and reduces DNA-binding of CLOCK-BMAL1 complex [18]. Despite 10%–20% of the rhythmic hepatic transcriptome is driven by liver intrinsic clocks [12,19], timing of food intake has actually been shown to be the

dominant driver for rhythmic liver transcriptome. A total of 70%–80% of the oscillating liver transcripts lose their rhythmicity when rhythmic feeding is disrupted in mice by fasting paradigms [20] or arrhythmic feeding [21]. Moreover, in mice with ablated clocks, time-restricted feeding can partially restore rhythmic liver transcriptome [19,20,22]. Despite that several nutrient-sensitive pathways, such as those involving CREB, AKT, mTOR, and ERK1/2, have been identified to modulate food-driven rhythmic transcripts [20,21], our understanding on the molecular mechanisms by which nutrient input regulate genome-wide rhythmic gene expression is far from complete.

O-linked-N-acetylglucosaminylation (O-GlcNAcylation) is a nutrient-sensitive post-translational modification (PTM) that modifies thousands of proteins and regulates nearly all aspects of cellular physiology, such as transcription, translation, cell signaling, immune response, and cell cycle [23–26]. O-GlcNAcylation is nutrient-sensitive given the substrate for O-GlcNAcylation, UDP-GlcNAc, is produced from the hexosamine biosynthetic pathway (HBP), which integrates glucose, amino acid, lipid and nucleotide metabolism [23–25]. Indeed, nutrient manipulation in cell media has been shown to alter cellular UDP-GlcNAc level and O-GlcNAcylation in several mammalian cell lines [27,28]. More importantly, Durgan and colleagues [29] and our previous study [30] showed that O-GlcNAcylation exhibits daily rhythmicity in mouse heart and *Drosophila* tissues, respectively, with the peak phase of O-GlcNAcylation rhythm closely aligned with timing of food intake. A large body of work have previously demonstrated the role of O-GlcNAcylation in regulating gene expression [23–25,31]. In the molecular clock, O-GlcNAcylation regulates the stability of transcription factors BMAL1 and CLOCK in a time-of-day manner [32]. Beyond the molecular clock, O-GlcNAcylation regulates gene expression by modifying the function of other gene-specific transcription factors [33–36], general transcriptional machineries [31,37–40], regulators of DNA methylation [41–44], histone modifiers and chromatin remodelers [45–49], and components of RNA spliceosome [50–52]. Therefore, we hypothesize that the hepatic nuclear proteome, especially proteins involved in orchestrating gene expression, could be rhythmically modified and regulated by O-GlcNAcylation.

In addition to directly modifying substrate proteins, O-GlcNAcylation also regulates protein functions by interacting with phosphorylation. O-GlcNAcylation and phosphorylation (G–P) can target the same serine/threonine residues on substrate proteins [23,27,53]. For example, in the molecular clock, competition of O-GlcNAcylation and phosphorylation at serine 662 (S662) residue of PER2 modulates transcriptional repressor activity of PER2 [54]. O-GlcNAcylation of histone H3 inhibits its mitosis-specific phosphorylation at S10, S28, and T32 to deter entry into mitosis [47]. A 5%−60% phosphoproteins have been shown to exhibit rhythmic phosphorylation over a 24-h period in liver tissue, and regulates many aspects of liver physiology, including gene expression, metabolism, cell signaling, apoptosis, and autophagy [55–58]. Since there is extensive interplay between O-GlcNAcylation and phosphorylation, it is possible that O-GlcNAcylation indirectly regulates rhythmic liver physiology through modulating daily phosphoproteome.

In this study, we showed that hepatic nuclear proteins exhibit daily oscillation of O-GlcNAcylation in liver of mice subjected to night-restricted feeding (NRF). We then performed O-GlcNAc chemoenzymatic labeling in combination with label-free mass spectrometry (MS) proteomics on mouse liver samples collected at 6 time-points over the day–night cycle to identify O-GlcNAcylated nuclear proteins and observed that proteins involved in transcriptional regulation were enriched in rhythmically O-GlcNAcylated proteins. To support our observation of time-of-day differences in protein O-GlcNAcylation status, we also performed site-specific O-GlcNAc analysis by tandem mass tag (TMT) quantitative proteomics at two time points (ZT11 and ZT23) over the day–night cycle and identified 125 O-GlcNAc sites with day–night differences belonging to 101 proteins. We then evaluated potential proteome-wide G–P crosstalk by performing TMT site-specific phosphoproteomic analysis using the same liver samples we used for global O-GlcNAc proteomics. Potential rhythmic G–P crosstalk was identified by close proximity of O-GlcNAc and phosphorylation sites as well as prediction by G–P interplay motif analysis. We performed functional analysis on one such G–P interplay motif in CLOCK, a key circadian transcriptional activator, and showed that G–P interplay regulates CLOCK transcriptional activity.

Finally, to determine if altering timing of food intake impacts daily O-GlcNAcylation rhythm, we compared O-GlcNAcylation patterns in liver of mice fed under NRF and day-restricted feeding (DRF). We observed dampening of

PLOS Biology

global O-GlcNAcylation rhythm of liver nuclear proteins in mice fed under DRF, while site-specific O-GlcNAcylation analysis revealed differential responses of O-GlcNAcylation sites when timing of food intake was altered. Our results provided evidence showing nutrient-sensitive O-GlcNAcylation regulates daily hepatic proteins involved in transcriptional regulation.

## Results

### Hepatic nuclear protein O-GlcNAcylation exhibits robust daily oscillation

We have previously reported that protein O-GlcNAcylation, a highly nutrient-sensitive PTM, exhibits daily rhythmicity in *Drosophila* tissues, and this rhythm is altered when we modified the timing of food intake [30]. Given the mammalian liver is a highly metabolically responsive tissue [11,13–15], we hypothesize that protein O-GlcNAcylation could exhibit daily rhythmicity in the liver and contribute to the regulation of daily rhythms in liver physiology. Liver samples from C57BL/6 male mice were collected every 4 h over a 24-h period (12 h:12 h light–dark (LD)). Since restricting timing of food intake to natural feeding window in *Drosophila* has been shown to enhance daily O-GlcNAcylation rhythm [30], we subjected mice to night-restricted feeding (NRF). To measure O-GlcNAcylation level, the O-GlcNAc moieties on nuclear protein were biotinylated using GalT1-dependent chemoenzymatic labeling and detected using streptavidin-HRP as previously described [30,59]. We only analyzed nuclear fractions, as our primary goal for this study was to investigate nutrient-sensitive PTM mechanisms that regulate the hepatic transcriptome. We observed that a group of nuclear proteins, ranging from 50 KD to 250 KD, are O-GlcNAcylated (Fig 1A) and global nuclear proteins exhibit rhythmic O-GlcNAcylation over a day–night cycle, as determined by Rhythmic Analysis Incorporating Nonparametric (RAIN) [60]. Nuclear protein O-GlcNAcylation peaks at zeitgeber time (ZT) 20 (calculated by RAIN), which is eight hours after the beginning of feeding period (Fig 1A and 1B). To confirm that the detected signals were indeed O-GlcNAcylated proteins, unlabeled samples were processed in parallel, where GalT1 labeling enzyme is absent. As compared to labeled samples (Lanes 3–4), almost no signal was detected in the unlabeled samples (Lanes 1–2), indicating that the signals observed in labeled samples are specific for O-GlcNAcylated proteins (S1 Fig).

It is possible that feeding–fasting cycles contribute to daily rhythmicity in protein O-GlcNAcylation by rhythmically supplying UDP-GlcNAc as substrate via hexosamine biosynthetic pathway (HBP). To investigate this possibility, we performed targeted metabolomics on HBP metabolites from the same mouse livers used in the analysis of O-GlcNAcylation shown in Fig 1A. We were able to detect UDP-GlcNAc, the immediate substrate of O-GlcNAcylation, despite many of the metabolites in HBP were not detected using our current method (S1 File). Similar to what we observed in flies [30], UDP-GlcNAc exhibited daily rhythm in mouse liver, with its peak preceding daily protein O-GlcNAcylation (UDP-GlcNAc: ZT12; O-GlcNAcylation: ZT20, RAIN). Furthermore, the timing of food intake occurred earlier than the rise of UDP-GlcNAc level (feeding: ZT12–24; UDP-GlcNAc: ZT 3–7) (Fig 1C). Although mice were allowed to feed from ZT12 to 24, the majority of feeding events are expected to occur earlier in the dark period [12,21] Our data suggest that feeding–fasting cycles could drive the oscillation of UDP-GlcNAc, which results in the oscillation of protein O-GlcNAcylation. The impact of meal timing on UDP-GlcNAc and O-GlcNAcylation rhythms will be evaluated in later sections of the study.

### Label-free global and TMT site-specific O-GlcNAc proteomics identify daily changes of nuclear protein O-GlcNAcylation

O-GlcNAcylation modifies thousands of proteins involved in many fundamental cellular processes [23–25,31]. As O-GlcNAcylation is shown to be rhythmic in liver nuclei, we hypothesize that O-GlcNAcylation regulates rhythmic liver function by modifying nuclear proteins in a time-of-day manner. To test our hypothesis, we performed daily label-free MS O-GlcNAc proteomics using liver samples from C57BL/6 male mice that were collected every 4 h over a 24-h period (12 h:12 h LD). Biotinylated O-GlcNAc nuclear proteins were enriched by affinity purification and then quantified using mass-spectrometry (S2 Fig). To filter out non-specific proteins, unlabeled samples were processed as described above and affinity purified (S2 Fig). To identify O-GlcNAcylated proteins, labeled and unlabeled samples were compared by

 

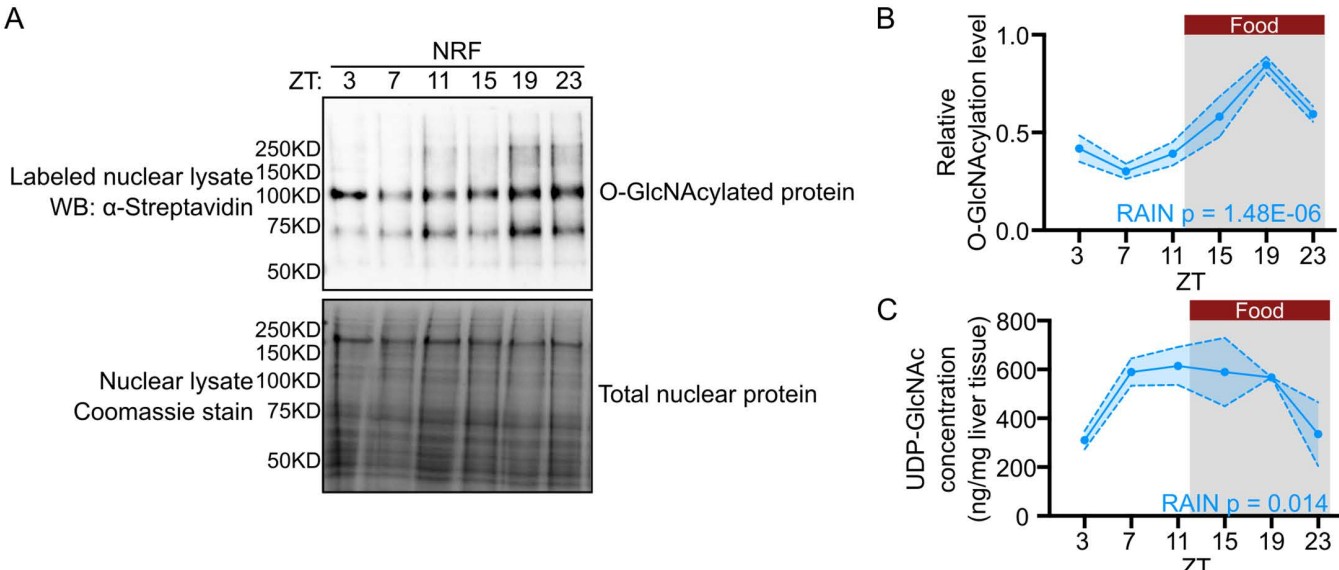

**Fig 1. Protein O-GlcNAcylation exhibits daily rhythmicity in mouse liver nuclei. (A)** Representative western blot showing the daily rhythm of nuclear protein O-GlcNAcylation in liver samples from mice subjected to night-restricted feeding (NRF). Total nuclear proteins were stained with Coomassie blue (bottom panel) and used for normalization. ZT, zeitgeber time (h). ZT0 denotes lights-on time and ZT12 denotes lights-off time. **(B)** Quantification of protein O-GlcNAcylation signals in panel **(A)** ($n = 3$). α-streptavidin signal of the whole lane was quantified **(C)** Daily rhythm of UDP-GlcNAc quantified using targeted metabolomics ($n = 3$). The grey shading in **(B)** and **(C)** illustrates the dark phase of the 24-h day–night cycle. The red bars indicate when food was available. ZT: zeitgeber time (h). Data are presented as mean ± SEM, and the blue shaded area represents SEM. *P* values indicate rhythmicity (RAIN). The numerical values of all replicates in **(B)** and **(C)** can be found in S1 Data.

Student *t* test with *p* value cutoff of 0.05. To identify proteins with rhythmic O-GlcNAcylation with normalization to their protein expression, we performed TMT proteomic analysis to obtain the expression level of each protein (see next section) (S2 and S3 Files, and S2 Fig). We identified 449 O-GlcNAcylated proteins in total. Multivariate PLS-DA analysis showed that O-GlcNAc proteome exhibits time-of-day variability (Fig 2A), which is further confirmed by pairwise multivariate analysis distinguishing each time point from the rest of the dataset (Fig 2B). Rhythmicity analysis was performed using JTK and RAIN with different *q*-value cutoffs ($q < 0.1$, 0.2, 0.3 or 0.4) (Fig 2C). O-GlcNAcylated proteins with RAIN $q < 0.2$ were used for further analysis, and the cutoff is chosen based on published -omics analysis of daily rhythms [61,62]. Most of the protein O-GlcNAcylation were found to occur during the dark phase when mice are active (Fig 2D and 2E). This is consistent with the overall O-GlcNAcylation profile measured by streptavidin-HRP on western blot (Fig 1A and 1B).

To identify the molecular pathways or processes that are rhythmically O-GlcNAcylated, we performed functional enrichment analysis on the proteins that we found to be rhythmically modified. We found that proteins related to gene expression were enriched (S3 Fig). To further dissect out the function of the rhythmically O-GlcNAcylated proteins, we classified these proteins into protein complexes using the CORUM database [63] (S4 File). We highlighted the protein complexes that regulate gene expression, such as transcriptional regulation, histone modification, chromatin remodeling, and RNA processing, and plotted their daily O-GlcNAcylation levels (S4 Fig). We also included several proteins that do not belong to specific protein complex in CORUM but were found to exhibit rhythmic O-GlcNAcylation. Consistent with previous studies [32], CLOCK is rhythmically O-GlcNAcylated with the peak time at the end of the feeding time window (S4 Fig). It is important to note that in addition to gene-specific transcription factor complexes (Bmal1-CLOCK-CRY1/2), general machineries that regulate gene expression are also rhythmically O-GlcNAcylated. These include general transcription factors (mediator and Brd4-TEFb complex), histone deacetylases (SP130), histone methyltransferases/demethylases (KDM6A, SETD1A and SETD2), chromatin remodelers (Zipzap/p200-Srf-p49/STRAP-Nkx2.5 and NoRC complex) and

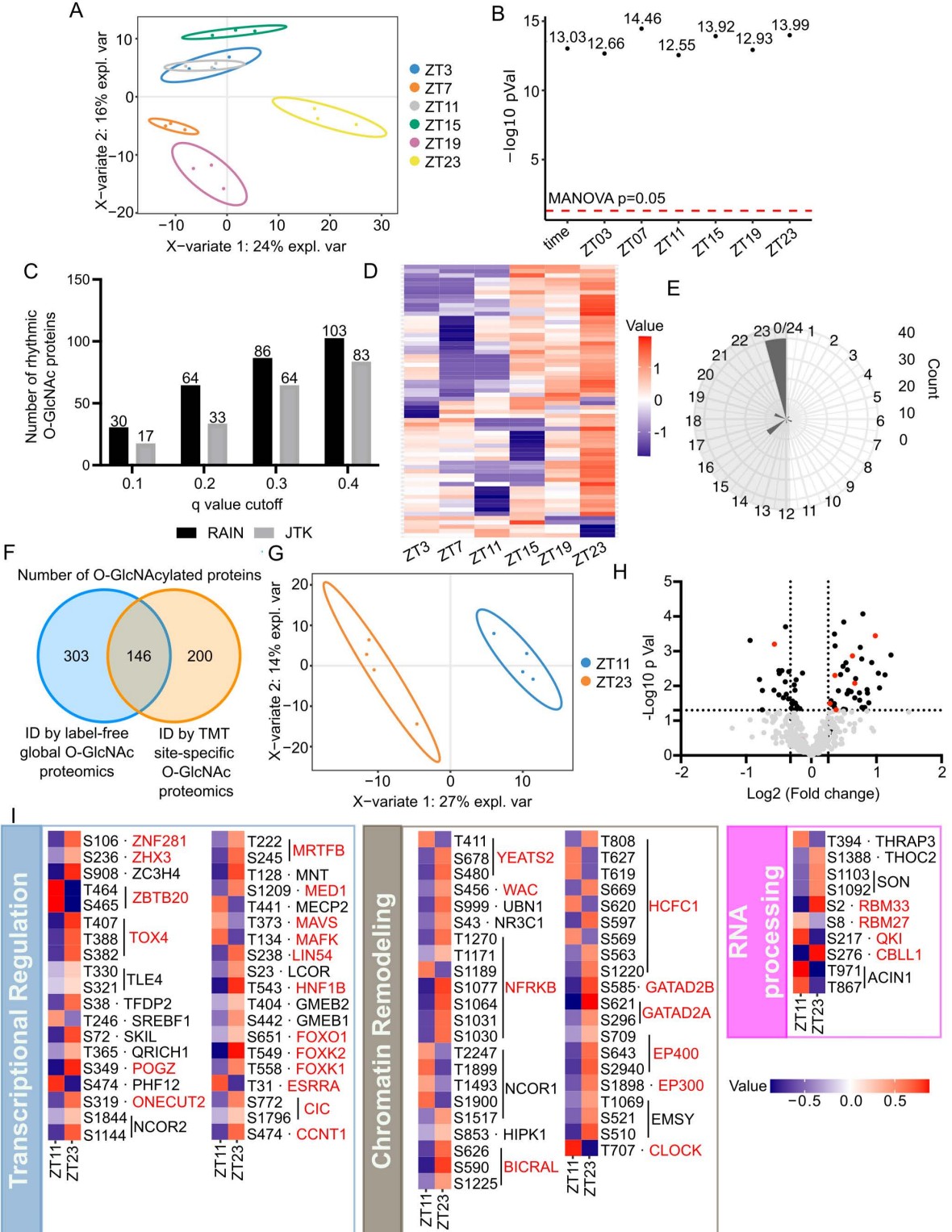

**Fig 2. Label-free global and TMT site-specific O-GlcNAc proteomics identify daily changes of nuclear protein O-GlcNAcylation. (A)** Score plot for PLS-DA model of label-free O-GlcNAc proteomics. **(B)** Negative log MANOVA *p* values for overall time and each time point tested in pairwise PLS-DA models. The red dash line represents MANOVA *p* value cutoff for significance. **(C)** The number of proteins with rhythmic O-GlcNAcylation detected by

RAIN and JTK using different cut-offs (*n* = 3). **(D)** Heat map showing the oscillation pattern of rhythmically O-GlcNAcylated proteins (*n* = 3). **(E)** Circular histogram showing the number of O-GlcNAcylated proteins peaking at each phase. The peak phase of O-GlcNAcylated proteins was calculated using RAIN. The gray shading illustrates the dark phase of a day–night cycle. **(F)** Venn diagram showing >40% O-GlcNAcylated proteins were identified in both global proteomics and site-specific O-GlcNAc proteomics. **(G)** Score plot for PLS-DA model of TMT site-specific O-GlcNAc proteomics. **(H)** Volcano plot displaying relative abundance of O-GlcNAcylation peptides between day and night. Horizontal dash line represents *p* < 0.05 cutoff (Student *t* test). Vertical dash lines represent fold change >20% (*n* = 4). **(I)** Heatmaps showing the relative O-GlcNAcylation level at indicated sites at ZT11 (day) and ZT23 (night) (*n* = 4). The proteins that were also identified as O-GlcNAc proteins in global proteomics are hightlighted in red.

RNA processing enzymes (FUS and RBM33) (S4 Fig). These data suggest rhythmic O-GlcNAcylation may modulate hepatic gene expression at multiple layers.

To gain further insights into the temporal functions of O-GlcNAcylation, we next performed quantitative site-specific analysis of protein O-GlcNAcylation. Based on our results thus far, we collected mouse liver at ZT11 and ZT23, two time-points that showed global differences in O-GlcNAcylation levels and are 12 h apart, and quantified site-specific O-GlcNAcylation using TMT-MS. In total, we identified 505 high-confidence O-GlcNAc sites on 346 proteins, with 338 sites (119 proteins) that were quantifiable (S5 File). Over 40% of O-GlcNAcylated proteins (146 proteins) identified by TMT-MS were also identified by label-free global O-GlcNAc proteomic analysis (Fig 2F). Multivariate PLS-DA analysis showed that O-GlcNAcylation sites quantified by TMT-MS exhibits day–night differences (Fig 2G). 125 O-GlcNAc sites on 101 proteins showed day–night O-GlcNAcylation differences (>20%-fold changes) (Fig 2H). Consistent with O-GlcNAcylation determined by Western blots (Fig 1A and 1B) and label-free global analysis (Fig 2D and 2E), over 60% of the 152 O-GlcNAc peptides quantified by TMT-MS display higher O-GlcNAcylation at nighttime (Fig 2H and 2I). Notably, we also found sites that exhibit higher O-GlcNAcylation at daytime. We reasoned these sites potentially belong to less abundant nuclear proteins that are not well represented in Western blots (Fig 1A and 1B), highlighting the importance of site-specific analysis.

We next performed functional enrichment analysis on the proteins harboring O-GlcNAcylation sites with day–night differences in site-specific analysis (S5 Fig). Consistent with enrichment analysis of rhythmic O-GlcNAcylation identified in label-free global analysis (S3 Fig), we found proteins involved in transcriptional regulation (Fig 2I). These include general transcriptional regulator (CCNT1, MED1, LIN54), proteins belonging to major chromatin remodeler subfamilies (GATAD2A, GATAD2B, BICRAL, NFRKB), and RNA processing enzymes involved in mRNA splicing and transport (THRAP3, SON, THOC2). Taken together, O-GlcNAcylation likely mediates the metabolic regulation of daily hepatic transcriptome via other mechanisms in addition to direct modulation of the molecular clock.

## TMT-MS profiles daily rhythm of nuclear phosphoproteome

Since both O-GlcNAcylation and phosphorylation modify serine/threonine residues on substrate proteins, many studies have been devoted to investigate G–P crosstalk [23,27,47,53,54,64,65]. Within a 24-h period, phosphoproteome has been shown to oscillate in diverse tissue types, including in mouse liver [55–57]. Therefore, we hypothesize that O-GlcNAcylation contributes to rhythmic liver physiology via daily G–P crosstalk. To test this hypothesis, we performed TMT-MS to analyze daily nuclear phosphoproteome using the same tissues we used for label-free global O-GlcNAc proteomics analysis described above (S2 Fig). We identified 19,593 phosphopeptides, belonging to 3,272 phosphoproteins (S6 File). To identify proteins with rhythmic phosphorylation independent of their protein expression, we performed a separate TMT proteomic analysis for normalization (S3 and S6 Files). Multivariate PLS-DA analysis showed that phosphoproteome exhibits time-of-day variability (Fig 3A), which is further confirmed by pairwise multivariate analysis distinguishing each time point from the rest of the dataset (Fig 3B). Rhythmicity analysis was performed using JTK and RAIN with different *q*-value cutoffs (*q* < 0.1, 0.2, 0.3 or 0.4) (Fig 3C). Phosphoproteins with RAIN *q* < 0.2 were used for further analysis. Compared with published circadian phosphoproteomics datasets [56,57], we identified 1820 additional rhythmic phosphosites (S6D Fig, S7 File, and S1 Table). Interestingly, we observed that most of the phosphorylation events happen during the dark phase, coinciding with the timing of O-GlcNAcylation (Fig 3D and 3E).

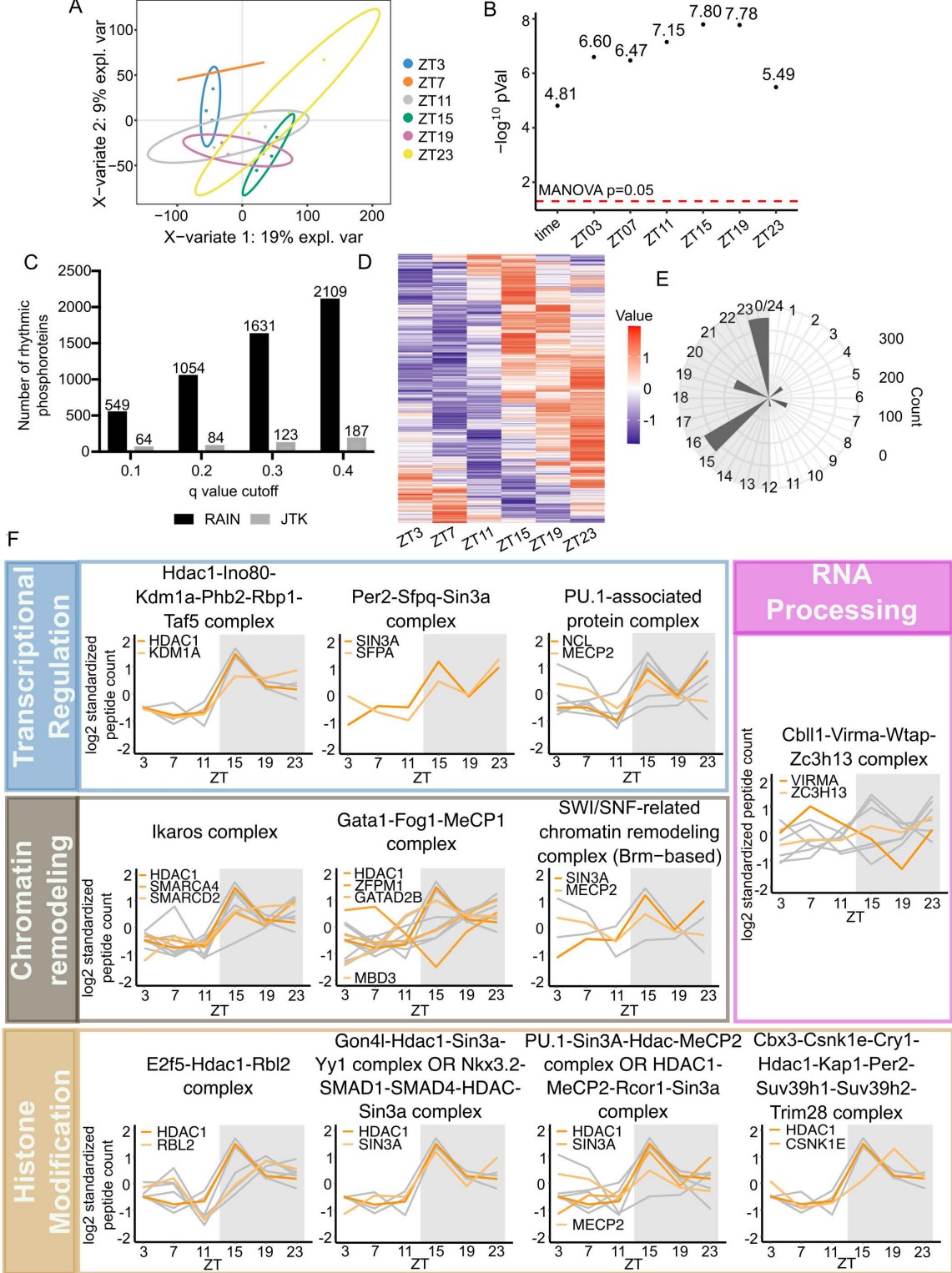

**Fig 3. Daily nuclear phosphoproteome of mouse liver from NRF mice. (A)** Score plot for PLS-DA model. **(B)** Negative log MANOVA *p* values for overall time and each time point tested in pairwise PLS-DA models. The red dash line is showing the MANOVA *p* value cutoff for significance. **(C)** The number of proteins with rhythmic O-GlcNAcylation detected by RAIN and JTK using different cut-offs (*n* = 3). **(D)** Heat map showing daily rhythmicity

of phosphopeptides ($n$=3). **(E)** Circular histogram showing the number of phosphopeptides peaking at each phase as determined by RAIN. The gray shading illustrates the dark phase of a day–night cycle. **(F)** Rhythmically phosphorylated proteins were classified into protein complexes based on known association and function, and the subunits with daily rhythmicity in phosphorylation were plotted in graphs. The protein complexes were further classified and grouped according to their biological functions as indicated on the left boxes. The gray lines indicate the oscillation of each phosphopeptide, while the orange lines indicate the average phosphorylation pattern of each protein subunit. Each gray line represents the average of three biological replicates. When only one peptide of individual protein was rhythmically phosphorylated (Per2-Sfpq-Sin3A complex panel), we used orange lines to represent phosphorylation status of individual peptides, hence there are no gray lines in that specific panel. The gray shading on each graph illustrates the dark phase of a day–night cycle.

To find out the kinases that can potentially rhythmically phosphorylate the hepatic nuclear proteome, PTMphinder [66] was used to enrich phospho-motifs of phosphopeptides peaking during dark phase (Fig 3E), and kinases were predicted using group-based phosphorylation site predicting and scoring (GPS) 5.0 [67]. To increase the confidence of our kinase prediction, we also selected out the kinases that are either rhythmically expressed, rhythmically O-GlcNAcylated, or rhythmically phosphorylated from our datasets (S8 File). We then classified these kinases by their phases calculated by RAIN. The circadian kinases were identified by selecting the common ones in the phase-matched rhythmic kinases in our proteomic datasets and the GPS 5.0-predicted kinase list. We were able to find several circadian kinases that were identified in previous circadian phosphoproteomic studies [56,57], such as AKT and CDK (S1 Table). We also identified several circadian kinases that although have not been found in previous circadian phosphoproteomics analysis, were shown to regulate circadian rhythm via targeted analysis, such as S6K [68] (S1 Table).

Functional enrichment analysis showed that rhythmic phosphoproteome are enriched in proteins involved in transcriptional regulation, RNA processing, and chromatin organization (S6 Fig). We then performed protein complex analysis to further identify the molecular pathways that are rhythmically phosphorylated (S9 File). Fig 3F illustrates protein complexes that are involved in transcriptional regulation and were found to contain multiple subunits that are rhythmically phosphorylated in our dataset. Consistent with previous findings, we identified a core clock protein complex, PER2-SFPQ-SIN3A complexes (Fig 3F). Additionally, we identified several general transcriptional complexes (Hdac1-Ino80-Kdm1a-Phb2-Rbp1-Taf5 and PU.1-associated complexes), chromatin remodeling complexes (Ikaros and SWI/SNF complexes), and RNA processing complexes (Cbll1-Virma-Wtap-Zc3h13 complexes) as rhythmically phosphorylated (Fig 3F). Among phosphorylation sites we identified in this analysis, some of them have previously been characterized with functions in gene expression regulation and beyond (S2 Table). For other sites with unknown functions, we identified the phosphorylation events locating in or adjacent to functional domains based on PhosphoSitePlus database [69] and these novel sites could have potential functions regulating liver physiology (S3 Table). Taken together, our data suggest that the daily rhythm of the phosphoproteome could regulate the rhythmic hepatic transcriptome.

### Identification of proteins that are modified by rhythmic O-GlcNAcylation and phosphorylation

O-GlcNAcylation can occur at the same residues or adjacent residues targeted by phosphorylation to regulate phosphorylation events [23,27,47,53,54,64,65]. Since both O-GlcNAcylation and phosphorylation were found to oscillate over a day–night cycle, we sought to identify rhythmic phosphorylation events that can be potentially regulated by daily O-GlcNAcylation rhythm of substrate proteins. By comparing results of our label-free global O-GlcNAc and TMT site-specific phosphoproteomic analysis, we found 14 proteins exhibiting both rhythmic O-GlcNAcylation and phosphorylation (Fig 4A). Comparison between the phases of rhythmic global O-GlcNAcylation and site-specific phosphorylation on the same proteins showed that many O-GlcNAcylation and phosphorylation rhythms are phase locked (Fig 4A), suggesting the two PTMs potentially facilitate each other. However, some proteins exhibit out-of-phase (phase-shifted) relationships in global O-GlcNAcylation and site-specific phosphorylation (Fig 4A), suggesting these two PTMs could potentially inhibit each other in these cases. Visualizing this data in circular density plot also showed that most of the O-GlcNAcylation

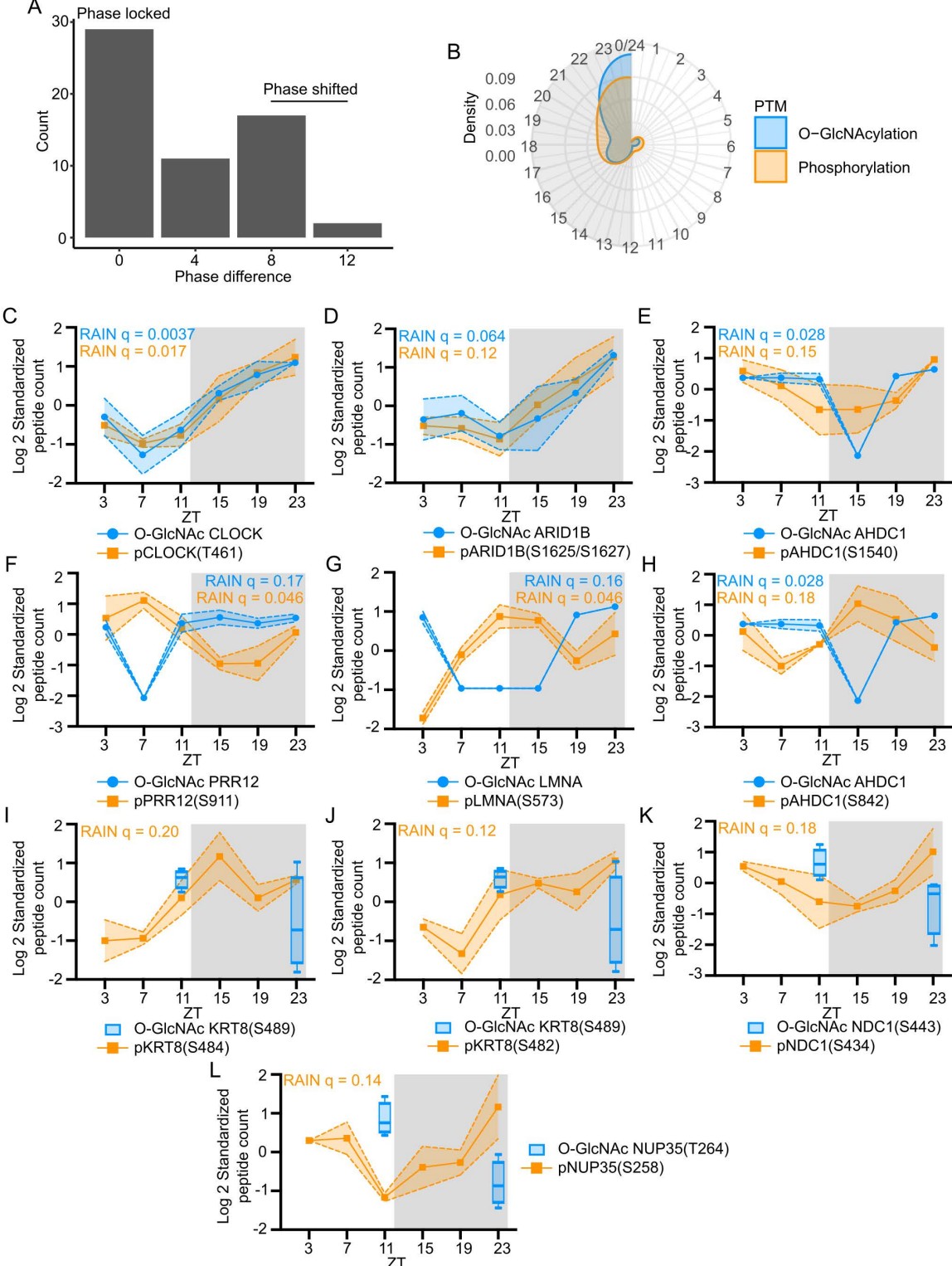

**Fig 4. Interaction between the O-GlcNAc proteome and phosphoproteome over a day–night cycle. (A)** Histogram to illustrate the phase difference between rhythmic O-GlcNAcylation and phosphorylation. Fifty-eight phosphopeptides in 14 (phosphorylated and O-GlcNAcylation) proteins were analyzed. Phase of site-specific phosphorylation rhythms and global O-GlcNAcylation rhythms were calculated by RAIN and phase difference ≤ 2 h was

defined as phase locked and phase difference ≥ 8 h as phase shifted [128,129]. **(B)** Circular density plot showing the peak phase of O-GlcNAcylation and phosphorylation events on proteins that exhibit rhythmicity in both modifications. The gray shading illustrates the dark phase of a day–night cycle. **(C–H)** Examples of global O-GlcNAcylation compared to the temporal phosphorylation profiles on the same proteins (RAIN: $q<0.2$, $q$ values were indicated). CLOCK and ARID1B represent proteins with in-phase rhythmic O-GlcNAcylation and phosphorylation **(C and D)**, while PRR12 and LMNA represent proteins with out-of-phase rhythmic O-GlcNAcylation and phosphorylation **(F and G)**. AHDC1 represent proteins with both in-phase **(E)** and out-of-phase **(H)** rhythmic O-GlcNAcylation and phosphorylation. Data are presented as mean ± SEM ($n=3$), and the shaded area around the curve represents SEM. **(I–L)** Examples of site-specific O-GlcNAcylation (at ZT11 and ZT23) identified by TMT site-specific O-GlcNAc proteomics ($n=4$) compared to the temporal phosphorylation profiles ($n=3$) on the indicated sites at 6 time-points (RAIN: $q<0.2$; $q$ values were indicated). O-GlcNAcylation with >20%-fold changes at ZT11 and ZT23 were defined as having day–night differences [124,125]. The gray shading illustrates the dark phase of a day–night cycle. Data are presented as mean ± SEM, and the shaded area around the curve represents SEM.

and phosphorylation occur within the same time window over the 24-h day–night cycle (Fig 4B). Analysis on individual phosphopeptide showed that some phosphorylation happens in phase with O-GlcNAcylation, such as pCLOCK(T461) (Fig 4C) and pARID1B(S1625/S1627) (Fig 4D), while other phosphorylation events are out of phase with O-GlcNAcylation, as exemplified by pPRR12(S911) (Fig 4F) and pLMNA(S573) (Fig 4G). pAHDC1(S1540), and pAHDC1(S842) demonstrate the situation that depending on the specific sites on even a single substrate protein, O-GlcNAcylation and phosphorylation can be in phase (Fig 4E) or out of phase (Fig 4H). Additionally, protein complex analysis identified 3 protein complexes out of these 14 proteins with both rhythmic O-GlcNAcylation and phosphorylation, including transcription factor complex (e.g., Arntl-Clock-Cry1/2 complex) and chromatin remodeling complex (e.g., 1-Fog1-MeCP1 complex).

In addition to the 14 proteins with both oscillating O-GlcNAcylation and phosphorylation, we expect that O-GlcNAc sites on more proteins could locate in G–P interplay motifs, given that the overall O-GlcNAcylation status of proteins detected in our label-free global O-GlcNAcylation analysis may underestimate potential rhythmicity of individual O-GlcNAc sites. To encourage future studies on daily G–P crosstalk, we took a subset of rhythmically phosphorylated proteins that are also O-GlcNAcylated in our label-free global O-GlcNAc proteomic dataset, and searched whether these rhythmic phosphosites belong to any G–P interplay motifs (S10 File). We found 167 phosphosites within G–P motifs on 134 proteins (S10 File). These 134 proteins (containing the 167 sites) include components in general transcription machinery (MED7, PCF11, and TCEA3, etc.), gene-specific transcription factors (NFIL3, SOX6 DBP, RBL2, etc.), histone modification (KAT5, KMT2D, and GATAD2A), chromatin remodeling (SMARCC2), RNA processing (SRSF5, CWC25 and FIP1), and nucleo-cytoplasmic transportation (POM121 and NUP98/153/214).

Finally, to further predict potential G–P crosstalk, we compared results of TMT site-specific O-GlcNAc proteomics with phosphoproteomics. We found 109 pairs of O-GlcNAc sites and phosphosites in close proximity (within 10 amino acids), indicating potential G–P crosstalks (S10 File). Among them, 82 showed day–night differences of either O-GlcNAcylation or phosphorylation. We found a number of potential G–P crosstalk proteins harboring sites with day–night differences in both O-GlcNAcylation and phosphorylation, such as in KRT8, NDC1, and NUP35 (Fig 4I–4L). Of these, phosphorylation levels at NDC1(S434) and NUP35(T258) exhibit negative correlation with O-GlcNAcylation level at adjacent sites (Fig 4K and 4L). In conclusion, our data suggest that G–P crosstalk has the potential to contribute to regulation of rhythmic protein function in mouse liver.

## Daily G–P crosstalk on CLOCK

To gain a deeper understanding of the potential crosstalk between O-GlcNAcylation and phosphorylation over a 24-h period, we asked whether proteins with rhythmic phosphorylation and O-GlcNAcylation identified in our studies harbor previously identified G–P interplay motifs [64,65]. We found 11 phosphorylation sites that harbor three different G–P interplay motifs (Fig 5A), where O-GlcNAcylation and phosphorylation either compete for the same modification site or have inhibitory effects when in close proximity. Among these 11 phosphorylation sites, the function of CLOCK(S431) has been characterized.

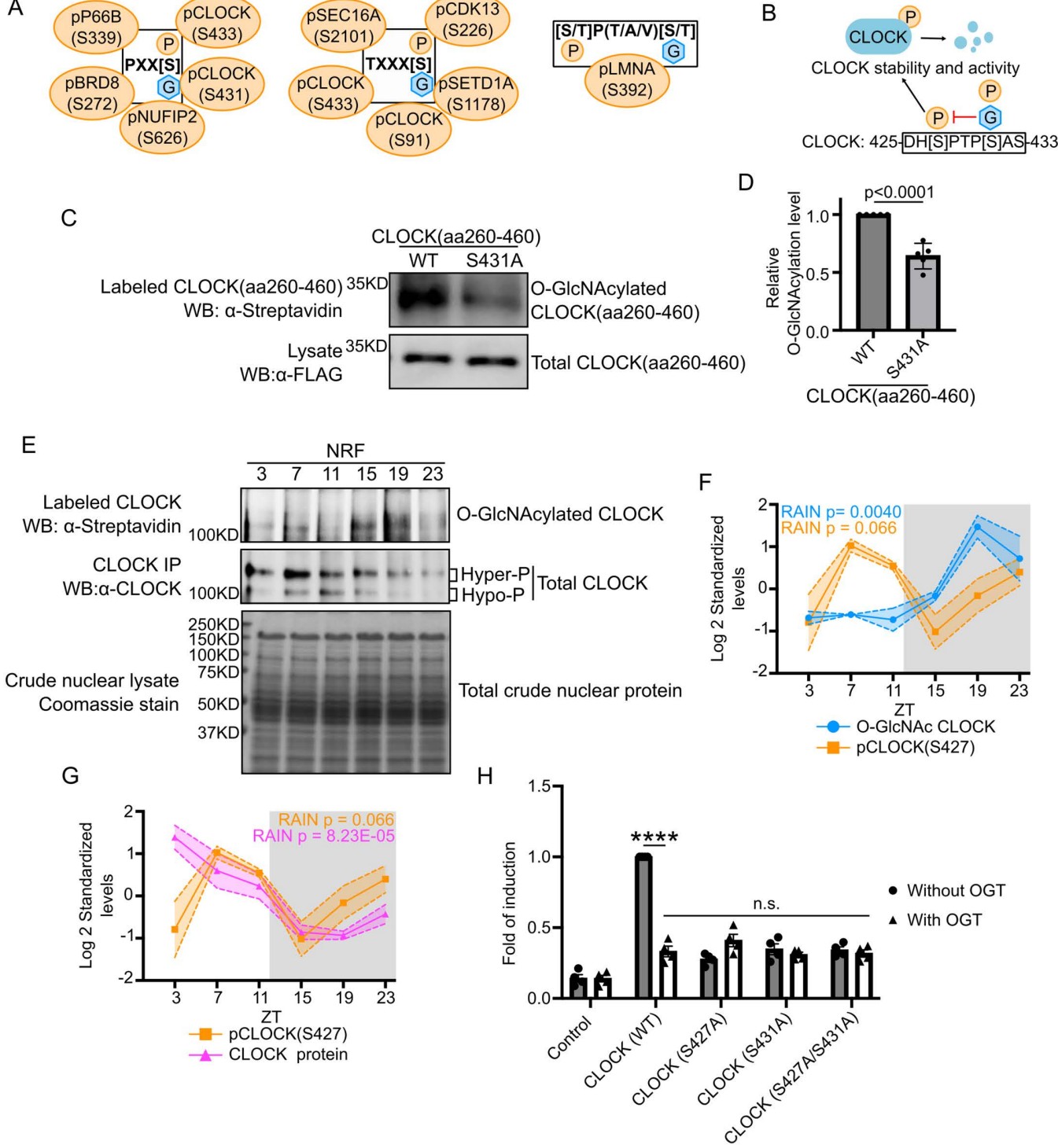

**Fig 5. O-GlcNAcylation interacts with phosphorylation to regulate rhythmic function of CLOCK. (A)** Schematics showing different types of O-GlcNAcylation and phosphorylation interplay motifs. Phosphorylation sites located within these motifs are indicated in the bubbles around each motif. O-GlcNAcylation either targets the same sites or approximal sites as indicated in each motif. **(B)** Model showing competition between O-GlcNAcylation and phosphorylation at CLOCK(S431) to inhibit CLOCK(S427) phosphorylation. Since S431 and S427 belong to a phospho-degron, O-GlcNAcylation at S431 could stabilize CLOCK by reducing phosphorylation at both sites and affect the transcriptional activity of CLOCK. **(C)** Representative blot showing

that CLOCK S431A mutation reduced the O-GlcNAcylation level of CLOCK(aa260–460). **(D)** Quantification of protein O-GlcNAcylation signals in panel **(C)** ($n = 5$). Student $t$ test $p$ value is indicated. **(E)** Representative blot showing the daily rhythms of CLOCK global O-GlcNAcylation (Top panel) and protein level (Middle panel). Total nuclear proteins were stained with Coomassie blue (Bottom panel) and used for normalization. Mice were subjected to NRF. **(F)** Graph comparing the daily patterns of CLOCK global O-GlcNAcylation vs. CLOCK(S427) phosphorylation levels (RAIN: $q = 0.496$; one-way ANOVA $p = 0.036$). **(G)** Graph comparing the daily patterns of CLOCK(S427) phosphorylation (RAIN: $q = 0.496$; one-way ANOVA $p = 0.036$) vs. CLOCK protein levels. The gray shading illustrates the dark phase of a day–night cycle. Data are presented as mean ± SEM ($n = 3$), and the shading area around the curves represents SEM. **(H)** Luciferase reporter assay performed in HEK293T cells showing that OGT and CLOCK mutations affect the transcriptional activity of CLOCK. **$p < 0.0021$, ****$p < 0.0001$, one-way ANOVA with post-hoc Dunnett's test. The numerical values of all replicates in **(D)**, **(F)**, **(G)**, and **(H)** can be found in S1 Data.

CLOCK(S431) is the priming phosphorylation site for glycogen synthase kinase-3 (GSK3) [70]. Once CLOCK(S431) is phosphorylated, GSK3 phosphorylates S427, which promotes the degradation of CLOCK. Given global O-GlcNAcylation of CLOCK has been shown to increase its stability [32], we hypothesize that O-GlcNAcylation stabilizes CLOCK in a time-of-day manner by competing with S431 phosphorylation to inhibit S427 phosphorylation (Fig 5B). To assess whether CLOCK(S431) is a O-GlcNAc site, we performed O-GlcNAc chemoenzymatic labeling followed by Western blotting in HEK293T cells expressing either CLOCK(WT) or CLOCK(S431A). A small fragment of CLOCK (aa260–460) was used in this experiment to better observe O-GlcNAcylation within this region. We observed that CLOCK(S431A) variant exhibits decreased O-GlcNAcylation as compared to CLOCK(WT), suggesting that CLOCK(S431) is indeed O-GlcNAcylated (Fig 5C and 5D). To validate the daily CLOCK O-GlcNAcylation rhythm detected in our label-free global O-GlcNAc proteomics analysis, we applied targeted chemoenzymatic labeling following immunoprecipitation of CLOCK. Consistent with the results from global O-GlcNAc analysis, CLOCK O-GlcNAcylation exhibits robust daily rhythmicity (Figs 5E, 5F, and S7A). CLOCK was detected in immunocomplexes of α-CLOCK bound beads, but not empty beads, demonstrating the specificity CLOCK immunoprecipitation (S8A Fig).

Although we detected S431 phosphorylation on multiple peptides in our dataset, the S431 phosphorylation always co-occurs with nearby phosphorylation events, making it difficult to quantify the actual level of S431 phosphorylation. We instead used S427 phosphorylation as the functional readout for the competition between O-GlcNAcylation and phosphorylation at S431. S427 phosphorylation displays daily changes ($p = 0.0036$; one-way ANOVA), despite not rhythmic defined by RAIN ($p = 0.066$) (Fig 5F). When we overlaid CLOCK O-GlcNAcylation rhythm in mouse liver from targeted analysis with the daily phosphorylation profile of S427, we observed out of phase relationship (peak O-GlcNAcylation: ZT20; peak pS427: ZT8, RAIN). Between ZT7 and ZT11, when global CLOCK O-GlcNAcylation level is low, S427 phosphorylation level remains high. The opposite is observed during the night period (ZT19) (Fig 5F). These observations suggest that CLOCK O-GlcNAcylation and S427 phosphorylation do not co-occur and CLOCK O-GlcNAcylation likely inhibits S427 phosphorylation.

Since CLOCK(S427/S431) belongs to a phospho-degron [70], we assayed CLOCK protein level in mouse liver over a 24-h cycle using Western blots (Fig 5E and 5G). The daily CLOCK protein levels detected in our proteomic dataset and Western blots have similar trends (S7B Fig). We then compared levels of daily S427 phosphorylation and CLOCK protein (Fig 5G). As S427 phospho-occupancy rises at ZT3 and peaks at ZT8 (RAIN), CLOCK protein level decreases starting ZT3 (Fig 5G). Between ZT15 and ZT23, CLOCK phosphorylation level rises again, while CLOCK protein level stays low. All these observations support the conclusion that daily changes of S427 phosphorylation contributes to the oscillation of CLOCK protein abundance.

We then sought to investigate whether O-GlcNAcylation at S431 affects CLOCK transcriptional activity since it has been established in the literature that CLOCK transcriptional activity is tightly linked to its stability [70–73]. Specifically, less stable CLOCK has higher transcriptional activity. This may seem counterintuitive but it has been established that qualitative changes in highly unstable CLOCK promote its transcriptional activity (also seen in many other transcription factors, reviewed in [74,75]). We assayed CLOCK transcriptional activity in HEK293T cells using a *per2-luciferase*

(*per2-luc*) reporter assay [76]. We compared *per2-luc* expression in cells transfected with CLOCK(WT), S427A, S431A, or the double non-phosphorylatable mutant S427/431A in the presence or absence of OGT (Fig 5H). To compensate for the fact the CLOCK mutants affect CLOCK protein level, we adjusted the amount of CLOCK plasmids for transfection to obtain an equal amount of CLOCK expression in each group of cells (S8B Fig). As expected, we observed CLOCK(WT) increases *per2-luc* signal and this signal is significantly reduced when CLOCK(WT) is coexpressed with OGT. This is consistent with previous findings that CLOCK O-GlcNAcylation increases its stability [32]. We then tested the single non-phosphorylatable mutants (S427A and S431A) and double mutant (S427/431A) and observed that they all exhibited lower transcriptional activity when compared to CLOCK(WT), and coexpression of OGT did not impact *per2-luc* activity (Fig 5H). Taken together, these data suggest the crosstalk between O-GlcNAcylation and phosphorylation at S431 modulates CLOCK transcriptional activity through regulating S427 phosphorylation. Future experiments using pS431 and pS427 phospho-specific antibodies in the presence/absence of OGT could provide further support for this model.

### Food intake at daytime in mice disrupts protein O-GlcNAcylation rhythm

We previously showed that O-GlcNAcylation is regulated by both feeding–fasting cycles and the molecular clock in flies [30]. We hypothesize that timing of food intake could affect daily O-GlcNAcylation rhythm in mouse liver. To test this hypothesis, we subjected mice to day-restricted feeding (DRF) and compared their nuclear protein O-GlcNAcylation pattern with mice subjected to NRF (Fig 6A). Mice treated with DRF lost their daily O-GlcNAcylation rhythm, and the global O-GlcNAcylation level of hepatic nuclear proteins remained at a relatively high level through the day–night cycle (Fig 6B and 6C). We recognize analysis of global O-GlcNAc level may be more representative of the status of abundant nuclear proteins. To evaluate potential differential responses of individual O-GlcNAcylation sites to changes in feeding time, we performed site-specific O-GlcNAc TMT proteomic analysis of liver nuclear proteins extracted from mice subjected to DRF (Fig 6D and 6E) in parallel to samples from NRF mice presented earlier (Fig 2F–2I). Consistent with global nuclear protein O-GlcNAcylation detected by Western blots (Fig 6B and 6C), DRF leads to reduced number of sites with day–night differences (>20%-fold change) in O-GlcNAcylation (from 128 down to 100 sites, or 101 to 74 proteins) (Fig 6D). Notably, O-GlcNAcylation levels in the majority of detected peptides are inverted under DRF (Fig 6E), while a small fraction of peptides showed unaltered day–night O-GlcNAcylation patterns or reduced day–night differences. Functional enrichment analysis on the O-GlcNAc peptides with inverted day–night profiles in NRF versus DRF liver resulted in proteins involved in transcriptional regulation, chromatin remodeling, and RNA processing (Fig 6F). These data suggest the dysregulation of nuclear protein O-GlcNAcylation may contribute to the disruption in liver transcriptome previously observed in DRF condition.

To investigate whether the altered O-GlcNAcylation pattern in DRF mice (Fig 6A–6F) results from altered daily UDP-GlcNAc rhythm, we used targeted metabolomics to detect UDP-GlcNAc levels in livers of mice subjected to either NRF or DRF. Our results showed that DRF liver exhibited dampened daily UDP-GlcNAc rhythmicity and the level of UDP-GlcNAc remained at a higher level overall when compared to that in NRF liver (Fig 6G). This suggests altered UDP-GlcNAc level likely contributed to altered O-GlcNAcylation in DRF mice liver (Fig 6G), although other yet established mechanisms are also involved given the various patterns of disruptions in O-GlcNAcylation rhythms in mice subjected to DRF. To rule out the possibility that the altered UDP-GlcNAc levels in DRF mouse liver is primarily due to increased food intake, we measured daily total food consumption of NRF and DRF mice (Fig 6H). We did not observe a significant increase in food consumption in the DRF group. DRF group instead fed slightly less than NRF group, indicating that food consumption is not the major cause of increased UDP-GlcNAc level. In conclusion, the effect of timing of food intake on O-GlcNAcylation rhythm in mouse liver is associated with altered UDP-GlcNAc profile.

### Discussion

We investigated the contribution of nutrient-sensitive O-GlcNAcylation as a post-translational mechanism to regulate the daily function of proteins involved in hepatic gene expression. Despite a substantial body of work showing that the

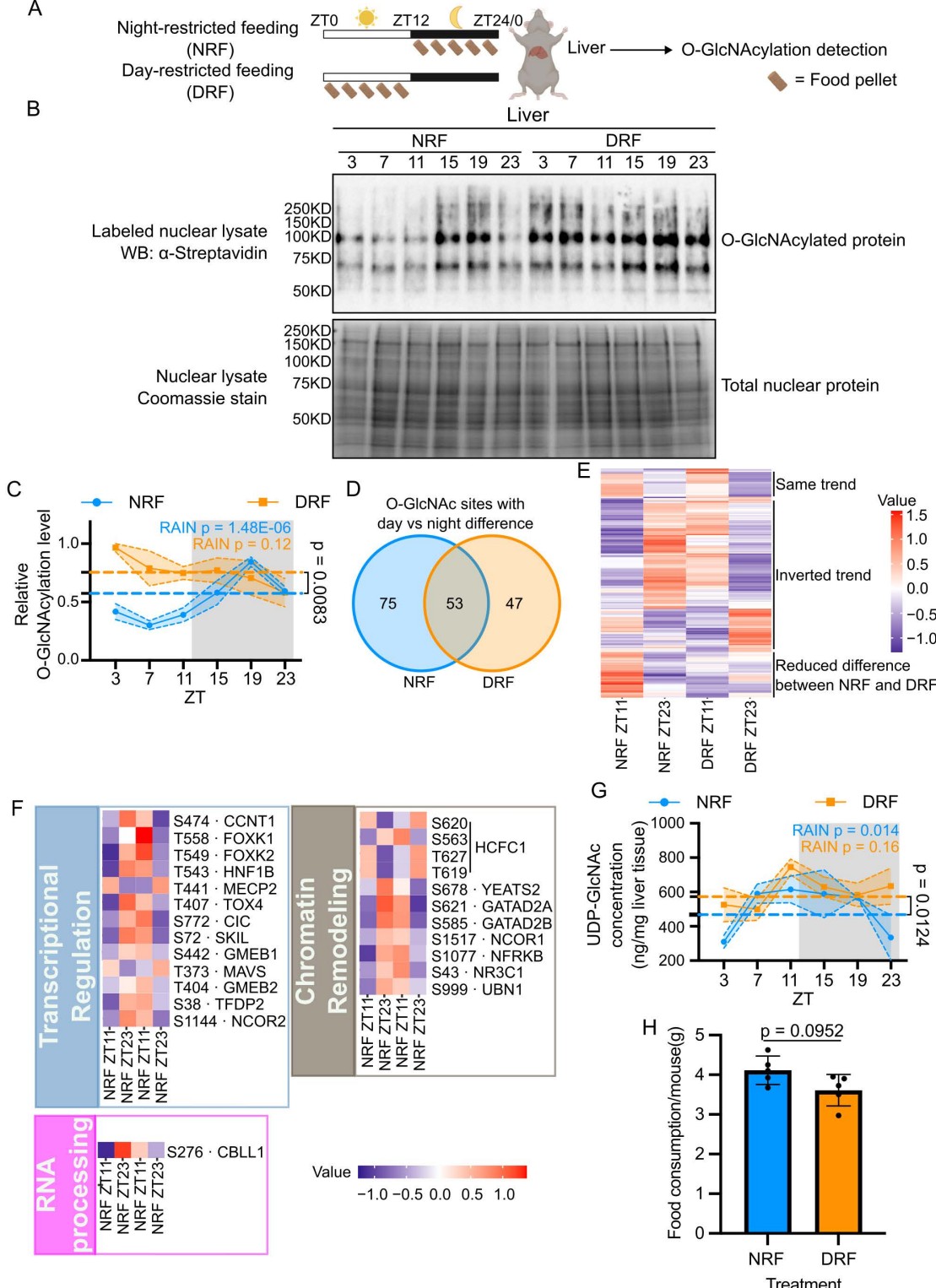

**Fig 6. Time of feeding affects daily protein O-GlcNAcylation rhythm in mouse liver. (A)** Schematics showing the study design. An 8-week-old C57BL/6 male mice were subjected to either NRF or DRF. The NRF group was only fed during the dark phase (ZT12–24), while the DRF group was only fed during the light phase (ZT0–12). After 3 weeks of treatment, mouse livers were collected at 4 h intervals over one day, and global O-GlcNAcylation

of liver nuclear proteins were detected. **(B)** Representative blot showing the daily rhythm of nuclear protein O-GlcNAcylation in liver samples from the NRF vs. DRF mice (Top panel). Total nuclear proteins were stained with Coomassie blue (Bottom panel) and used for normalization. **(C)** Quantification of protein O-GlcNAcylation signals in **(B)** ($n = 3$). The α-streptavidin signal of the entire lane was quantified. The NRF data is replotted from Fig 1B. **(D)** Venn diagram to compare the O-GlcNAc sites with day vs. night difference (>20% fold change) in NRF and DRF liver. **(E–F)** Heatmaps showing the relative abundance of site-specific O-GlcNAcylation between ZT11 (day) and ZT23 (night). Heatmap is generated using detected O-GlcNAcylated peptides in **(E)** and O-GlcNAc sites in **(F)**. **(G)** Daily rhythm of UDP-GlcNAc were quantified using targeted metabolomics ($n = 3$). The NRF data is replotted from Fig 1C. The dashed lines represent the median levels of O-GlcNAcylation **(C)** or UDP-GlcNAc **(G)** in NRF and DRF groups, and Student $t$ test was used to assess significant difference. The gray shading illustrates the dark phase of a day–night cycle. Data are presented as mean ± SEM, and the shading area around the curves represents SEM. **(H)** Food consumption of NRF and DRF mice over a 24-h period ($n = 5$). $p = 0.0952$, Mann–Whitney test. The numerical values of all replicates in **(C)**, **(G)**, and **(H)** can be found in S1 Data.

rhythmic hepatic transcriptome is highly responsive to the timing of food intake [11,14,15,19–22], our understanding of the mechanisms by which rhythmic feeding–fasting activity regulates genome-wide gene expression is far from complete. Here, we showed that nuclear proteins in mouse liver exhibit rhythmic O-GlcNAcylation over a 24-h day–night cycle when mice were fed at night (Fig 1). This is supported by both global and site-specific O-GlcNAcylation analysis. Proteins involved in transcriptional regulation, histone modification, chromatin remodeling and RNA processing are enriched among proteins with temporal changes of O-GlcNAcylation (Figs 2, S3, and S5). Combining results from targeted metabolomics (Figs 1C and 6G), phosphoproteomics (Fig 3), biochemistry (Figs 1A, 1B, 5C– 5H, and 6A–6C) and G–P interplay analysis (Figs 4, 5A, and 5B), we formulated a model illustrating the role of O-GlcNAcylation in mediating the effect of metabolic input on the daily function of nuclear proteins in liver (Fig 7). We propose that environmental signaling via the molecular clock and the timing of metabolic input from food intake converge to regulate daily rhythm in the O-GlcNAc proteome. O-GlcNAcylation targeting the same site or proximal site has the potential to affect phosphorylation, as exemplified by CLOCK. Therefore, O-GlcNAcylation can either directly modulate proteins involved in regulation of gene expression or in an indirect manner by interacting with phosphorylation of these proteins.

To better understand the function of O-GlcNAcylation on hepatic gene expression, we identified O-GlcNAcylated proteins using both label-free global analysis and TMT site-specific analysis (Fig 2). There was substantial overlap between the O-GlcNAcylated proteins identified using the two methods, and these proteins were enriched for transcriptional regulators. Previous studies showed that O-GlcNAcylation modulates functional outcomes of several transcription factors detected in our rhythmic O-GlcNAc proteome, including EWS, MEF2D, and GATA4 [77–80]. O-GlcNAcylation increases the transcriptional activity and stability of GATA4 [79,80], while O-GlcNAcylation of MEF2D reduces insulin secretion in pancreatic β-cells [78]. Therefore, it is possible that the time-of-day specific functions of these proteins could be driven by rhythmic O-GlcNAcylation. We anticipate that technologies in quantitative O-GlcNAc proteomics will continue to advance and further reveal the prevalent nature of O-GlcNAcylation in transcriptional regulators, similar to that of phosphorylation [81–83]. Future functional characterization studies are warranted to characterize the function of O-GlcNAc sites with temporal differences identified in current study to understand how they regulate daily rhythms in hepatic gene expression.

Since there exists crosstalk between O-GlcNAcylation and phosphorylation at the same or adjacent site on its protein substrate, we investigated the possibility that rhythmic O-GlcNAcylation can interact with the phosphoproteome to regulate hepatic nuclear proteins (Figs 4 and 5). Given that many of the 14 proteins with rhythmic O-GlcNAcylation and phosphorylation are known to play important roles in various cellular processes, such as transcriptional regulation (AHDC1 and TIF1A), histone modification (KANSL1 and SETD1A), RNA processing (CDK13), and chromatin modification (ARID1B, BRD8, and P66B), it is important to understand how daily G–P crosstalk could modulate the activities of these proteins to regulate daily hepatic physiology. Additionally, a broader motif search on O-GlcNAcylated proteins with rhythmic phosphorylation predicted that 167 rhythmic phosphosites could potentially be regulated by G–P interplay motifs (S10 File). Fast evolving MS-based quantitative O-GlcNAc profiling [84–88] will continue to provide more insights into

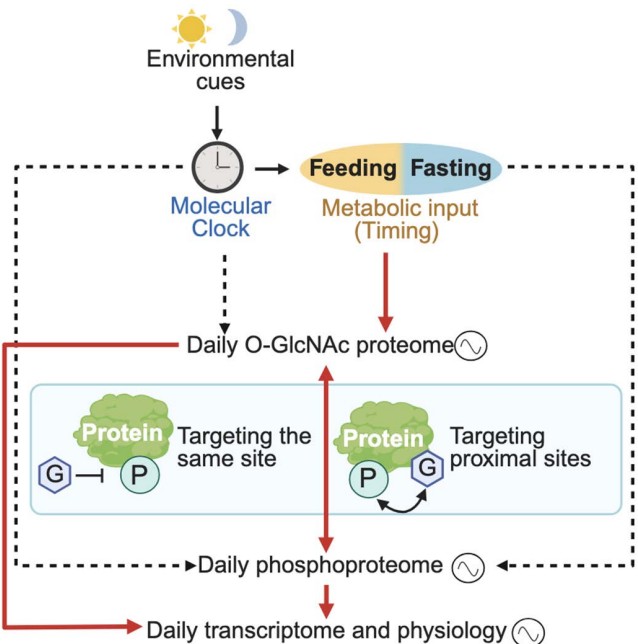

**Fig 7. Model describing how daily rhythmic O-GlcNAc proteome integrates environmental and metabolic signals to regulate transcriptomic and physiological rhythms.** Environmental cues signaling through the molecular clock and metabolic cues from feeding–fasting cycles regulate daily rhythms in the O-GlcNAc proteome. Rhythmic O-GlcNAcylation can directly shape daily rhythmicity in transcriptome and physiology. In addition, rhythmic O-GlcNAcylation can also interact with rhythmic phosphorylation. Different forms of crosstalk between O-GlcNAc proteome and phosphoproteome are shown in the middle box. The red solid arrows indicate the regulations described in this study, while the dashed arrow indicates regulations discovered by previous studies [30,57,58]. Figure created using BioRender.com, licensed to the lab of J.C.C.

G–P interplay. Indeed, in addition to G–P interplay at CLOCK phospho-degron (Fig 5), our TMT site-specific O-GlcNAc proteomic revealed O-GlcNAcylation at CLOCK(S450), proximal to S441 and S446 phosphorylation sites. How individual O-GlcNAcylation sites collectively regulate CLOCK remains to be investigated. Recently, CLOCK has been shown to be phosphorylated in response to osmotic stress [89] and is modified by another nutrient-sensitive PTM, S-palmitoylation [90]. How O-GlcNAcylation interacts with other PTMs in response to complex inputs to regulate daily physiology also remains an open question.

Our results showed that O-GlcNAcylation is rhythmic in mouse liver with a peak phase that aligns with the window of food intake (Fig 1). Given the nutrient-sensitive nature of O-GlcNAcylation, it is perhaps not surprising to observe similar phenomenon in mouse heart [29] and fly body [30]. Therefore, O-GlcNAcylation may represent a conserved mechanism by which nutrient input regulates daily protein functions under natural feeding conditions. We observed that DRF dampens the global O-GlcNAcylation rhythm of hepatic nuclear proteins, instead of shifting the phase of O-GlcNAcylation to align with the altered feeding period (Fig 6). However, it is clear from site-specific analysis of O-GlcNAcylated proteins extracted from DRF versus NRF mice that the situation is more complex and different O-GlcNAcylation sites respond differently when timing of food intake is altered. Whereas some O-GlcNAcylation sites similarly show reduced day–night differences, similar to what we observed in the global O-GlcNAcylation analysis, some O-GlcNAcylation sites exhibited inverted day–night patterns between DRF and NRF. O-GlcNAcylation of these proteins may therefore contribute to the DRF-induced phase shift of many transcripts observed in previous studies [11,14,15,91–93]. Future site-specific and functional investigations will be essential to reveal how O-GlcNAcylation of these proteins regulate their target genes in response to food intake.

It is important to note that there are other mechanisms involved in regulating daily rhythms of hepatic nuclear protein function in addition to O-GlcNAcylation. These additional mechanisms could include O-GlcNAcylation of proteins in other subcellular compartments. O-GlcNAcylation of cytoplasmic proteins are also likely to oscillate. O-GlcNAcylation of cytoplasmic proteins can regulate biological processes such as translation and immune response [23,26], nuclear shuttling of proteins [94–96], and relay signals into nucleus via signaling pathways [20,21,97,98]. How compartmentalization of UDP-GlcNAc [80] regulates daily O-GlcNAcylation of nuclear and cytoplasmic proteins remains unknown. Despite our understanding of O-GlcNAcylation regulation of nutrient-sensitive signaling pathways, such as those involving mTOR [99], AMPK [100,101], and insulin [102], whether and how O-GlcNAcylation interacts with these nutrient-sensitive signaling pathways and other nutrient-sensitive PTMs in a time-of-day manner remains to be investigated. Moreover, we observed that DRF can alter daily rhythms of bile acids and S-adenosylmethionine (SAM) and likely downstream signaling pathways (S9 Fig). Bile acid can regulate gene expression through a nuclear receptor, farnesoid X receptor (FXR) [16,103], whereas SAM is the substrate for methylation of DNA, RNA and proteins [104–107]. Therefore, a comprehensive understanding of how the hepatic transcriptome responds to timing of food consumption will require an integration of nuclear and cytoplasmic protein O-GlcNAcylation, and other nutrient-sensitive mechanisms.

In summary, we showed that O-GlcNAcylation exhibits daily oscillation in hepatic nuclear proteins in mice and subjecting mice to DRF disrupts daily O-GlcNAcylation rhythm. Using proteomic approaches, we identified nuclear proteins exhibiting rhythmic O-GlcNAcylation and phosphorylation and evaluated potential G–P crosstalk at the proteomic level over a day–night cycle. Finally, our findings on CLOCK, a protein in the core molecular clock loop, highlight the daily crosstalk between O-GlcNAcylation and phosphorylation and its functional outcomes. Our results provide new insights into metabolic regulation of daily biological rhythms at the PTM level and shed light on the deleterious effects of improper mealtimes.

## Materials and methods

### Ethics statement

All experiments including animal feeding, treatment, and tissue collection were approved by the Institutional Animal Care and Use Committees (IACUC) of the University of California, Davis according to the National Institutes of Health and Animal Research: Reporting of In Vivo Experiments guidelines. The approval (protocol number: 24217) was granted by John Tupin, the director of institutional review board administration.

### Animal treatments and collection of liver samples

C57BL/6 male mice were ordered from JAX laboratory at the age of 6 weeks. Mice were first entrained for 2 weeks *ad libitum* in 12 h light:12 h dark cycle. Mice were then randomized into two time-restricted feeding (TRF) groups: night-restricted feeding (NRF) between ZT12 and ZT24 (ZT, zeitgeber time; ZT0 indicates light on, while ZT12 indicates light off) and day-restricted feeding (DRF) between ZT0 and ZT12. Normal chow food was provided in the above time-restricted feeding treatments and food intake was measured by comparing the food weight before and after each feeding window. The mice were TRF-treated for 3 weeks. Three weeks is enough time for DRF group to adapt to the new feeding window and fed comparable amount as NRF group [108]. It also allows the entrainment of the hepatic clock to be complete [14,15] and the rhythmic transcriptome would be altered [11]. After TRF treatment, mice were sacrificed every 4 h over a 24-h period and liver samples were collected. Samples were then flash frozen in liquid nitrogen, ground into powders and stored at −80°C until sample processing.

### Extraction of mouse liver nuclei proteins

Mouse liver nuclei were extracted according to Nagata and colleagues [109] with the following modifications. Liver samples were dounced in Buffer A (250 mM sucrose, 10 mM Tris HCl pH 7.5, 5 mM $MgCl_2$, 100 μM PUGNAc, and Roche EDTA-free protease inhibitor cocktail) with a loose pestle. The suspension was passed through a cell strainer (70 μm pore

size, Fisher Scientific, Hampton, NH) by centrifuging at 300 × g for 1 min at 4°C. The crude nuclei were pelleted by centrifugation at 600 × g for 15 mins at 4°C. To further purify the crude nuclei, the pellets were resuspended in 9 volumes of ice-cold Buffer B (2 M sucrose, 1 mM MgCl₂, and 10 mM Tris HCl pH 7.5) and were then centrifuged at 16000 × g for 15 mins. The supernatant was discarded and the pure nuclear pellets were resuspended in boiling buffer (100 mM Tris pH 8.0, 2% SDS, 100 μM PUGNAc, and Roche EDTA-free protease inhibitor cocktail) [59]. Nuclear proteins were extracted by heating at 95°C for 5 min. After centrifuging at 16000 × g for 5 mins to separate the soluble and non-soluble nuclear fractions, the nuclear protein lysate was transferred to a new tube.

## Expression and purification of bovine β-1,4-galactosyltransferase 1 (GalT1 Y289L)

GalT1 (Y289L), the chemoenzymatic labeling enzyme for detecting O-GlcNAcylated proteins, was expressed and purified according to Thompson and colleagues [59] The bovine GalT1 was obtained from Genomics-online (Cat#: ABIN3462403) and GalT1 tyrosine 289 is mutated into leucine to increase its activity for the chemoenzymatic labeling reaction [110,111]. GalT1(Y289L) was subcloned into pET22-6xHis vector and transformed into BL21(DE3) competent cells for expression of the GalT1(Y289L) protein. The harvested cells were lysed in PBS containing 1mM EDTA by sonication. After centrifuging at 14,000 × g for 30 min, the pellet was collected given GalT1(Y289L) is insoluble in PBS. The pellet was suspended in wash buffer (1× PBS, 25% sucrose, 1 mM EDTA, and 0.1% Triton X-100) and spun down; this was repeated 5 times. The pellet was then resuspended in PBS containing 1 mM EDTA and this was repeated twice to remove detergent. After the last centrifuge step, the pellet was resuspended in denaturing buffer (5 M guanidine hydrochloride, 0.3 M sodium sulfite) and treated with S-sulfonating agent, 2-Nitro-5-thiosulfobenzoate (NTSB). Ice-cold water was added to precipitate the GalT1(Y289L) protein and the mixture was centrifuged at 10,000 × g for 10 min. This step was repeated 3 times to remove residual NTSB. The final pellet was resuspended in 5 M guanidine hydrochloride and diluted 10 times in refolding buffer (0.5 M L-arginine, 50 mM Tris pH 8.0, 5 mM EDTA, 4 mM cysteamine, and 2 mM cystaminea). The resulting GalT1(Y289L) solution was dialyzed in dialysis buffer (50 mM Tris pH 8.0, 5 mM EDTA, 4 mM cysteamine, and 2 mM cystamine) and concentrated to 2 mg/ml.

## Chemoenzymatic labeling of O-GlcNAcylated proteins

Chemoenzymatic labeling of O-GlcNAcylated nuclear proteins was performed according to Thompson and colleagues [59] with the following modifications. The nuclear protein lysate was quantified and 100 μg protein was used for labeling. Protein lysate was treated with 25 mM DTT at 95°C for 5 mins and 50 mM iodoacetamide at room temperature for 1 h in the dark. Proteins were precipitated using chloroform–methanol method and dissolved in 40 μl 1% SDS with 20 mM HEPES pH 7.9 by heating up at 95°C for 5 mins. The O-GlcNAc group was first labeled with azide in the labeling buffer (20 mM HEPES pH 7.9, 50 mM NaCl, 2% IGEPAL-CA630, 25 μM PUGNAc, 25 mM NaF, 0.5 mM PMSF, 5.5 mM MnCl₂, 25 μM UDP-GalNAz, 50 μg/ml GalT1(Y289L), and Roche EDTA-free protease inhibitor cocktail). To remove the labeling reagents, proteins were precipitated using chloroform–methanol method and dissolved in 37.5 μl 1% SDS with 20 mM HEPES pH 7.9. Biotin was attached to the azide group in the biotinylation buffer (1× PBS, 100 μM biotin alkyne, 2 mM sodium ascorbate, 100 μM Tris (benzyltriazolylmethyl) amine, and 1 mM CuSO₄). To prepare for western blot after O-GlcNAc labeling, samples were precipitated and dissolved in 50 μl 1% SDS with 20 mM HEPES pH 7.9. O-GlcNAc proteins were detected using α-streptavidin-HRP (Cell Signaling Technologies, Danvers, MA) (1:10000). For background deduction, unlabeled samples were processed in parallel with GalT1(Y289L) excluded in the labeling buffer.

## Targeted gas chromatography coupled to time-of-flight (GC-TOF) mass spectrometry (MS) analysis and HILIC-TripleTOF MS/MS analysis

An amount of 10 mg liver samples were analyzed using GC-TOF MS and HILIC-TripleTOF MS/MS approaches as previously described [30,112]. Samples were extracted with acetonitrile/isopropanol/water 3:3:2 (v/v/v). For GC-TOF MS

analysis, 0.5 µl samples were injected into an Agilent 6890 gas chromatograph (Agilent, Santa Clara, CA) and analyzed using a Leco Pegasus IV time of flight mass (TOF) spectrometer. Result files were processed using the metabolomics BinBase database. For HILIC-TripleTOF MS/MS, 5 µl samples were injected onto a Waters Acquity UPLC BEH Amide column (Waters, Milford, MA) and analyzed using a 1×6,600 TripleTOF high resolution mass spectrometer (SCIEX, Framingham, MA). Mass spectra were analyzed and interpreted using MS-DIAL v3.90 [113]. Peak heights from both analyses were reported as quantification. Standard compounds from the hexosamine biosynthesis pathway (HBP) were diluted into multiple concentrations and analyzed in parallel to generate standard curves. The concentration of each detectable HBP metabolites in liver samples was then calculated based on the peak height and standard curves.

### Sample preparation for label-free global O-GlcNAc proteomics

Nuclear proteins were chemoenzymatically labeled as described above with 1 mg protein per samples. After the final precipitation, samples were dissolved in 400 µl 1% SDS with 50 mM Tris pH 7.5. A 600 µl supplementary buffer (50 mM Tris pH 7.5, 375 mM NaCl, and 2.5% IGEPAL-CA630) was added to dilute protein into immunoprecipitation (IP) buffer (50 mM Tris pH 7.5, 150 mM NaCl, 0.4% SDS, and 1% IGEPAL-CA630). The protein sample is quantified using Qubit before IP. O-GlcNAc proteins were immunoprecipitated using Dynabeads MyOne Streptavidin T1 (Fisher Scientific) and the IP reactions were incubated at room temperature for 2 h. The beads were washed with wash buffer (50 mM Tris pH 7.5, 400 mM NaCl, 1% IGEPAL-CA630, and 0.4% SDS) for 5 times and 50 mM $NH_4HCO_3$ for 8 times. After removing all the solutions through magnetic separation, beads were stored at −80°C prior to mass spectrometry analysis. To identify non-specific binding, unlabeled samples were processed in parallel with GalT1(Y289L) excluded in the labeling buffer.

### Label-free MS to identify O-GlcNAc proteins

The bead-bound proteins were digested on beads according to Mohammed and colleagues [114]. Briefly, samples were digested using 10 µl (10 ng/µl) of trypsin in 100 mM $NH_4HCO_3$ overnight at 37°C. Another 10 µl (10 ng/µl) of trypsin in 100 mM $NH_4HCO_3$ was added the next day to further digest for 4 h. The digested peptides were extracted twice with 5% formic acid and 60% acetonitrile.

Samples were analyzed by liquid chromatography–mass spectrometry (LC–MS) according to Sharma and colleagues [115]. The Nano-LC–MSMS was performed using a Dionex rapid-separation liquid chromatography system interfaced with an Eclipse (Fisher Scientific). Samples were loaded onto a fused silica trap column (Acclaim PepMap 100, 75 µm×2 cm, Fisher Scientific) and washed for 5 min at 5 µl/min with 0.1% TFA. An analytical column (Nanoease MZ peptide BEH C18, 130A, 1.7 µm, 75 µm×250 mm, Waters, Milfored, MA) for LC-MS/MS was brought in-line with the trap column. A segmented linear gradient was used to fractionate peptides at 300 nl/min: 4%–15% solution B in 30 min (solution A: 0.2% formic acid, and solution B: 0.16% formic acid, 80% acetonitrile), 15%–25% solution B in 40 min, 25%–50% solution B in 44 min, and 50%–90% solution B in 11 min. Before the next run started, solution B should return to 4% for 5 min. Data-dependent acquisition procedure was applied to acquire the spectrometry data. A cyclic series of a full scan were performed with a resolution of 120,000. The 20 most intense ions were analyzed by MS/MS (higher-energy collisional dissociation (HCD), relative collision energy 27%) with a dynamic exclusion duration of 20 s.

LC–MS/MS data were analyzed with Thermo Proteome Discoverer 2.4 (Fisher Scientific) into MASCOT Generic Format (MGF). The resulting peak list was searched against Uniprot mouse database and a database consisted of common lab contaminants (CRAP) using in house version of X!Tandem (GPM Fury [116]). During searching, fragment mass error was set at 20 ppm and parent mass error was set at ±7 ppm. Carbamidomethylation on cysteine was set as fixed modification. In primary search, methionine monooxidation was set as variable modifications, while in refinement search, asparagine deamination, tryptophan oxidation and dioxidation, methionine dioxidation, and glutamine to pyro-glutamine were included. Protease was set as trypsin C-terminal of R/K unless followed by P with 1 miscut allowed during the preliminary

search and 5 miscuts allowed during refinement. For minimum acceptable peptide and protein expectation scores, $10^{-2}$ and $10^{-4}$ were applied, respectively. The overall peptide false-positive rate was set to 0.07% [117].

## Tandem mass tag (TMT) site-specific O-GlcNAc proteomics

Sample preparation for quantitative O-GlcNAc proteomics was performed according to a procedure described previously [88], with some modifications. In brief, proteins (1.25 mg for each condition) extracted from liver nuclear lysates were reduced with DTT and alkylated with IAA, followed by chloroform–methanol precipitation. Proteins were then dissolved in 400 μl 1% SDS with 20 mM HEPES pH 7.9 by heating up at 95°C for 5 mins. Chemoenzymatic labeling was initiated by adding 490 μl of $H_2O$, 800 μl of labeling buffer (50 mM HEPES pH 7.9, 125 mM NaCl, 5% NP-40), 105 μl of 100 mM $MnCl_2$, 100 μl of 0.5 mM UDP-GalNAz, 5 μg GalT1(Y289L), 0.5 μl of PNGase F, and 1 μl of FastAP Thermosensitive Alkaline Phosphatase (Fisher Scientific), and incubating at 4°C overnight. Proteins were chloroform–methanol precipitated and dissolved in 1 mL of 0.5% SDS with 20 mM HEPES pH 7.9 for click chemistry by adding 11 μl of 10 mM PC Biotin Alkyne (Vector Laboratories, Newark, CA), 33 μl of 10 mM $CuSO_4$-BTTAA (1:2, measured as $CuSO_4$, Vector Laboratories), and 5 μL of 50 mM sodium ascorbate, and incubating at room temperature for 2 h. After precipitation, proteins were resuspended and digested by 50 μg of trypsin in 1.6 M urea with 50 mM HEPES pH 7.9. Peptides were desalted and resuspended in 1.25 mL of 100 mM triethylammonium bicarbonate (TEAB) buffer pH 8.5 for TMTpro16 labeling (Fisher Scientific), with two kits used by following the manufacturer's protocol. Labeled samples were pooled at 1:1 ratio and incubated with 1.2 mL of high-capacity NeutrAvidin agarose slurry (Fisher Scientific) for 3 h. The conjugated peptides were released in 0.1% formic acid (FA) by irradiation at 365 nm for 1 h. Subsequently, the released peptides were fractionated by High pH Reversed-Phase Peptide Fractionation Kit (Fisher Scientific) following the manufacturer's protocol. Ten fractions were collected, dried down, and dissolved in 0.1% FA for nanoUPLC–MS/MS analysis.

The nanoUPLC–MS/MS analysis was performed on a nanoAcquity UPLC system (Waters, Milford, MA) coupled with an Orbitrap Fusion Lumos mass spectrometer (Fisher Scientific), similar as described previously [88,118]. In brief, samples were loaded onto a C18 Trap column (Waters Acquity UPLC M-Class Trap, Symmetry C18, 100 Å, 5 μm, 180 μm × 20 mm) at 10 μL/min for 4 min. Separation was carried out on an analytical column (Waters Acquity UPLC M-Class, Peptide BEH C18, 300 Å, 1.7 μm, 75 μm × 150 mm) at a column temperature of 45°C with a flow rate of 400 nL/min. A 300-min gradient was used with buffer A (2% ACN, 0.1% formic acid) and buffer B (0.1% formic acid in ACN) as follows: 1% B at 0 min, 5% at 1 min, 30% at 210 min, 45% at 260 min, 90% at 270 min, 90% at 290 min, 1% at 290.1 min, and 1% at 300 min. The MS data were acquired in data dependent acquisition (DDA) mode by Orbitrap Fusion Lumos mass spectrometer with an ion spray voltage of 2.4 kV and an ion transfer temperature of 275°C. The mass spectra were recorded with Xcalibur 4.7. The MS parameters were set as below: Detector Type: Orbitrap; Orbitrap Resolution: 120,000; Scan Range: 350–1,800 m/z; RF Lens: 30%; AGC Target: Standard; Maximum Injection Time Mode: Auto; Microscans: 1. Charge state: 2–8; Cycle Time: 3 s. HCD product-dependent EThcD (HCD-pd-EThcD) with a dynamic exclusion duration of 40 s was applied for MS/MS acquisition. EThcD was triggered by the oxonium ions of HexNAc (m/z 126.055, 138.055, 144.066, 168.065, 186.076, and 204.086), as well as the major fragments resulting from the PC-biotin alkyne (m/z 300.130 and 503.210) observed in HCD scans. MS/MS parameters were set as below: Isolation Mode: Quadrupole; Isolation Window: 0.7 m/z; HCD collision energy: 30%. Detector Type: Orbitrap; Resolution: 50,000; AGC Target: Standard. Supplemental activation (SA) collision energy of EThcD was set as 30%.

Raw data were analyzed with Proteome Discoverer 2.4 (Fisher Scientific) with the Sequest search engine against Uniprot mouse database (TaxID: 10,090, downloaded on February 2, 2023; 17,137 sequences). The MS mass tolerance was set at ±10 ppm, MS/MS mass tolerance were set at ±0.02 Da. TMTpro on K and N-terminus of peptides and carbamidomethyl on cysteine was set as static modification. Mass addition of 502.202 Da (AMTzHexNAc2) on Ser/Thr/Tyr/Asn were set as dynamic modification, as well as asparagine deamidation, methionine oxidation, protein N-terminal acetylation, N-terminal methionine loss or N-terminal methionine loss plus acetylation. Protease was set as trypsin digest with 2

mis-cleavages allowed. Percolator was used for results validation. Concatenated reverse database was used for target-Decoy strategy. High confidence for protein and peptides was defined as FDR<0.01, medium confidence was defined as FDR<0.05.

For reporter ion quantification, HCD was selected as the activation type. S/N was used for reporter quantification peaks if all spectrum files have S/N values. Otherwise, intensities were used. Quantification value was corrected for isotopic impurity of reporter ions. Protein abundance of each channel was calculated using summed S/N of all unique + razor peptides with AMTzHexNAc2 on Ser/Thr/Tyr. Co-isolation threshold was set at 50%. Average reporter S/N threshold was set at 10. SPS mass matches percentage threshold was set at 65%.

## Tandem mass tag (TMT) site-specific phosphoproteomics

For parallel phosphoproteomic analysis using the same mouse livers used for label-free global O-GlcNAc proteomics analysis, mouse liver nuclei were extracted as described above. The nuclear pellets were resuspended in 2× laemmli buffer with 50 mM DTT. Nuclei were sonicated, followed by incubating at 95°C for 10 min. After centrifugation at 25,000 × $g$ for10 min at 4°C, the soluble fraction was saved in a new tube, while the insoluble fraction was further extracted using 8 M Urea. The insoluble fraction was centrifuged again, and the supernatant was combined with the soluble fraction. Protein concentration was determined by Pierce 660 nm Protein Assay Kit (Fisher Scientific). A 300 μg of lysate were run into SDS-PAGE for TMT-MS analysis. We chose to use TMT labeling method to increase the comparability of samples [119]. However, as the currently available TMT labeling method can maximumly label 18 samples (TMTpro-18plex) in one run, we were only able to process 3 biological replicates of mouse livers [119].

The gel slices were incubated with 10 mM DTT at 60°C for 30 min, and then with 20 mM iodoacetamide at room temperature for 1 h in dark. The samples were digested by trypsin at 1:50 (w:w, trypsin: sample) and incubated at 37°C overnight. The digested peptides were extracted and dried under vacuum and washed with 50% acetonitrile to pH neutral. The digested peptides were labeled with Thermo TMT18plex (Lot# for TMTpro16: XA341491; Lot# for last two channels: XA338617) followed by the manufacture's protocol. Labeled samples were pooled at 1:1 ratio for a small volume and analyzed with LC–MS/MS to get normalization factor. The labeling efficiency is 97.9%. Based on the normalization factor, the samples were pooled together and desalted with SepPack tC18 columns (Varian, WAT054960).

The desalted samples were fractionated by high-pH RPLC chromatography using Agilent 1100 series. The samples were solubilized in 200 μl of 20 mM ammonium (pH 10), and injected onto an Xbridge column (Waters, C18 3.5 μm 2.1 × 150 mm) using a linear gradient of 1% B/min from 2%–45% of B (buffer A: 20 mM ammonium, pH 10, B: 20 mM ammonium in 90% acetonitrile, pH 10). UV 214 was monitored. About 1 min fractions were collected. A 5% of fractions 29–46 was saved for proteome analysis and the rest of all fractions were dried until there is around 50 μl left for each fraction. Then the fractions were combined and dry completely under speed vacuum for phosphopeptide enrichment.

IMAC enrichment of phosphopeptides was adapted from Mertins and colleagues [120] with modifications. Ion-chelated IMAC beads were prepared from Ni-NTA Superflow agarose beads (Qiagen, MA). Nickel ion was stripped with 50 mM EDTA and iron was chelated by passing the beads through aqueous solution of 200 mM $FeCl_3$ followed by three times of water wash and one time wash with binding buffer (40% acetonitrile, 1% formic acid). The combined basic RP fractions were solubilized in binding buffer and incubated with IMAC beads for 1 h. After three times of wash with binding buffer, phosphopeptides were eluted with 2× beads volume of 500 mM potassium hydrogen phosphate, pH 7.0 and the eluate was neutralized with 10% formic acid. The enriched phosphopeptides were further desalted by Empore 3M C18 (2,215) StageTip[2] prior to nanoLC–MS/MS analysis.

Nano-LC-MSMS was performed using a Dionex rapid-separation liquid chromatography system interfaced with an Eclipse (Fisher Scientific). The enriched phosphopeptides and selected desalted fractions 29–46 were loaded onto an Acclaim PepMap 100 trap column (75 μm × 2 cm, Fisher Scientific) and washed with Buffer A (0.1% trifluoroacetic acid) for 5 min with a flow rate of 5 μl/min. The trap was brought in-line with the nano analytical column (nanoEase, MZ peptide

BEH C18, 130A, 1.7 μm, 75 μm × 20 cm, Waters) with flow rate of 300 nL/min with a multistep gradient (4% to 15% buffer B [0.16% formic acid and 80% acetonitrile] in 20 min, then 15%–25% B in 40 min, followed by 25%–50% B in 30 min). The scan sequence began with an MS1 spectrum (Orbitrap analysis, resolution 120,000, scan range from 350 to 1,600 Th, automatic gain control (AGC) target 1E6, maximum injection time 100 ms). The top S (3 s) duty cycle scheme were used for determining number of MSMS performed for each cycle.

For proteomic samples, SPS method was used. Peptide ions were first collected for MS/MS by collision-induced dissociation (CID) and analyzed in quadrupole ion trap analysis (automatic gain control (AGC) 2E4, normalized collision energy (NCE) 35, maximum injection time 120 ms, and isolation window at 0.7). Following acquisition of each MS2 spectrum, 10 MS2 fragment ions were captured in the MS3 precursor population using isolation waveforms with multiple frequency notches. MS3 precursors were fragmented by HCD and analyzed using the Orbitrap (NCE 55, AGC 1.5E5, maximum injection time 150 ms, resolution was 50,000 at 400 Th).

For phosphopeptide, MS/MS method was used. Parent masses were isolated in the quadrupole with an isolation window of 0.7 m/z, AGC 1E5, and fragmented with HCD with a normalized collision energy of 34%. The fragments were scanned in Orbitrap with resolution of 50,000. The MS/MS scan ranges were determined by the charge state of the parent ion but lower limit was set at 110 amu.

LC–MS/MS data were analyzed with Proteome Discoverer 2.4 (Fisher Scientific) with sequent search engine against Uniprot mouse database and a database consisted of common lab contaminants. For proteome data, The MS mass tolerance was set at ±10 ppm, MS/MS mass tolerance were set at ±0.4 Da. TMTpro on K and N-terminus of peptides and carbamiodomethyl on cysteine was set as static modification. Methionine oxidation, protein N-terminal acetylation, N-terminal methionine loss or N-terminal methionine loss plus acetylation were set as dynamic modifications. Protease was set as trypsin digest with 2 miscuts allowed. For phosphopeptide enriched sample runs, The MS mass tolerance was set at ±10 ppm, the MS/MS mass tolerance were set at ±0.02 Da. TMTpro on K and N-terminus of peptides and carbamiodomethyl on cysteine was set as static modification. Serine, threonine, and tyrosine phosphorylation, methionine oxidation, protein N-terminal acetylation, N-terminal methionine loss or N-terminal methionine loss plus acetylation were set as dynamic modifications. Protease was set as trypsin digest with 2 miscuts allowed. Percolator was used for results validation. Concatenated reverse database was used for target-Decoy strategy. High confidence for protein and peptides was defined as FDR < 0.01, medium confidence was defined as FDR < 0.05.

For reporter ion quantification, reporter abundance was set to use signal/noise ratio (S/N) if all spectrum files have S/N values. Otherwise, intensities were used. Quan value was corrected for isotopic impurity of reporter ions. Co-isolation threshold was set at 75%. Average reporter S/N threshold was set at 10. SPS mass matches percentage threshold was set at 65%. Protein abundance of each channel was calculated using summed S/N of all unique + razor peptide. For proteome data, the abundance was further normalized to summed abundance value for each channel over all peptides identified within a file. The same normalization factors were also used to normalize phosphopeptide abundance. Following the analytic method of TMT phosphoproteomics dataset in Li and colleagues [121], each phosphopeptide was further normalized to corresponding master protein abundance value if the protein group also reported abundance value in proteome data. If the protein abundance value was missing, then the original abundance was used. Whether a specific phosphopeptide is normalized or not is indicated in S6 File under the "Normalized" column.

## CLOCK O-GlcNAcylation detection

Around 300−400 mg liver tissue was used for CLOCK O-GlcNAcylation detection. The protocol was adapted from Yoshitane and colleagues [122] and Cao and colleagues [123] with the following modifications. Crude nuclei were extracted as described above with modified Buffer A (10 mM HEPES pH 7.9, 10 mM KCl, 0.1 mM EDTA, 1 mM DTT, 1 mM PMSF, 50 mM NaF, 100 μM PUGNAc, and Roche EDTA-free protease inhibitor cocktail). Nuclear proteins were extracted using nuclear extraction buffer (NEB) (20 mM HEPES pH 7.9, 10% Glycerol, 400 mM NaCl, 1% Triton X-100, 0.1% sodium

deoxycholate, 1 mM EDTA, 5 mM MgCl$_2$, 1 mM DTT, 0.5 mM PMSF, 10 mM NaF, Roche PhosSTOP, 100 μM PUGNAc, and Roche EDTA-free protease inhibitor cocktail) by rotating at 4°C for 30 mins. The soluble and non-soluble nuclear fractions were separated by centrifuging at 16,000 × $g$ for 15 min. Prior to immunoprecipitation, samples were diluted using buffer D (20 mM HEPES, pH 7.9, 10% Glycerol, 1 mM EDTA, 1 mM DTT, 0.5 mM PMSF, 10 mM NaF, Roche PhosSTOP, 100 μM PUGNAc, and Roche EDTA-free protease inhibitor cocktail) at 3:5 ($V_{NEB}$:$V_D$). Immunoprecipitation was performed using 4 μl α-CLOCK (Cat# ab3517, Abcam, Boston, MA) in combination with 20 μl gammabind Sepharose beads (GE HealthCare, Chicago, IL). A 5 μl of beads was collected for detection of CLOCK protein, and western blot was performed with α-CLOCK (1:3000).

For O-GlcNAcylation detection, the rest of the beads was washed twice with wash buffer (20 mM HEPES, pH 7.9, 10% Glycerol, 150 mM NaCl, 1 mM EDTA, 1 mM DTT, 2 mM MgCl$_2$, 0.5 mM PMSF, 10 mM NaF, 100 μM PUGNAc, and Roche EDTA-free protease inhibitor cocktail) and twice with reaction buffer (20 mM HEPES pH 7.9, 50 mM NaCl, 25 mM NaF, 0.5 mM PMSF, 5 mM MnCl$_2$, 1 μM PUGNAc, and Roche EDTA-free protease inhibitor cocktail). After washing, 16 μl of reaction buffer, 2 μl of GalT1(Y289L) and 2 μl of 0.5 mM UDP-GalNAz were added and the mixture incubated at 4°C overnight. On the second day, the beads were washed twice with the reaction buffer to remove excess UDP-GalNAz, and 37.5 μl 1% SDS with 20 mM HEPES pH 7.9 was added to the resin. Finally, biotinylation was performed using a biotinylation buffer (1× PBS, 100 μM biotin alkyne, 2 mM sodium ascorbate, 100 μM Tris (benzyltriazolylmethyl) amine, and 1 mM CuSO$_4$). To prepare for western blot, samples were precipitated using the chloroform–methanol method and dissolved in 50 μl 1% SDS with 20 mM HEPES pH 7.9. O-GlcNAcylated proteins were detected using α-streptavidin-HRP (Cell Signaling Technologies) (1:5000). For detecting the O-GlcNAcylation of CLOCK(aa260–460) overexpressed in HEK293T cells, similar method was used except that cells were extracted using RIPA buffer supplemented with Roche PhosSTOP and 100 μM PUGNAc. Immunoprecipitation was performed using 1 μl α-FLAG (MilliporeSigma).

### Luciferase reporter assay to test CLOCK transcriptional activity

For luciferase assay, HEK293T cells were transfected with 100 ng *per2-luc*, 100 ng *Renilla-luc*, 50 ng pcDNA3.1-*Bmal1*-Flag, 50 ng pcDNA3.1-Myc-*Ogt* or 50 ng pcDNA3.1-Myc, 400 ng pcDNA3.1-*Clock*(WT)-Flag or 1,000 ng pcDNA3.1-*Clock*(X)-Flag, where X is either S427A, S431A or S427A/S431A. pcDNA3.1-*Clock*(WT)-Flag was a kind gift from Dr. Erquan Zhang and *Clock* mutants were generated by mutagenesis. Measurements were performed 24 h after transfection and Dual luciferase reporter assay kit (Vazyme, Nanjing, China) was used according to manufacturer's instructions.

### Bioinformatic and statistical analysis

For label-free MS O-GlcNAc proteomics, peptide count was first normalized to the input protein amount for IP with streptavidin beads. Similar to the TMT phosphoproteomics analysis, the O-GlcNAcylated protein peptide counts were further normalized to their protein level obtained from the TMT-MS analysis performed in parallel using the same samples, if the protein was detected [121]. If the protein expression level was not sufficiently resolved, the original peptide count was maintained [121]. Whether a specific O-GlcNAc protein is normalized or not is indicated in S2 File under the "Normalized" column. To identify O-GlcNAcylated proteins, labeled samples and unlabeled samples were compared using Student $t$ test with $p$ value cutoff of 0.05, and the unlabeled samples were used for background deduction ($n = 3$).

For TMT site-specific O-GlcNAc proteomics, the presence of day–night differences were defined as having differences between day versus night time-points larger than 20%. This criterion was adapted from published literatures on quantitative site-specific O-GlcNAc proteomics [124,125].

PLS-DA and MANOVA analysis were performed using mixOmics [126]. Rhythmicity, phase and period length of O-GlcNAc proteins and phosphopeptides were determined using Rhythmic Analysis Incorporating Nonparametric (RAIN) [60] and MetaCycle [127]. For kinase analysis, the kinase motifs on phosphopeptides were enriched using

PTMphinder [66] and the enriched motifs were then fed into group-based phosphorylation site predicting and scoring (GPS) 5.0 to predict the kinases [67]. We utilized the CORUM database [63] to identify protein complexes with rhythmic O-GlcNAcylation and/or phosphorylation on complex subunit(s). RAIN, MetaCycle, mixOmics, PTMphinder, and protein complex analysis were performed in R v4.2.2. The O-GlcNAcylation levels of CLOCK(aa260–460), and the median levels of O-GlcNAcylation and UDP-GlcNAc in DRF and NRF groups were compared using two-tailed unpaired Student $t$ test in GraphPad Prism 10.0 (GraphPad Software, La Jolla California USA), while the feeding amount between DRF and NRF was compared using Mann–Whitney test. For comparing the transcriptional activity of CLOCK(S427/S431) mutants (Fig 5), one-way ANOVA with post-hoc Dunnett's test was performed in Prism 10.0. For testing how CLOCK pS427 changes with time (Fig 5F and 5G), one-way ANOVA was also performed in Prism 10.0.

## Supporting information

**S1 Fig. Validation of chemoenzymatic labeling method for O-GlcNAcylation detection in mouse liver tissues.** Two samples for the NRF group were chosen to show time-specific difference of liver nuclear O-GlcNAcylation (Lanes 3–4). Lanes 1–2 show the unlabeled samples processed in parallel with the labeled samples in Lanes 3–4. GalT1 enzyme was excluded during the processing of unlabeled samples.
(TIFF)

**S2 Fig. Study design for O-GlcNAc proteomics and TMT phosphoproteomics analysis.** Flow chart illustrating the design of the proteomics experiments using mice subjected to NRF. Liver samples were collected at 6 time-points over a 24 h day–night cycle. Nuclear extracts were used for site-specific TMT phosphoproteomics and global O-GlcNAc proteomics. During the process of TMT phosphoproteomics, samples were first labeled with TMT tags, and the majority of the samples were used for enrichment of phospho-peptide followed by MS analysis. A small portion of TMT-labeled samples were used for proteomic analysis, where samples were directly subjected to MS analysis without enrichment. The protein levels obtained from proteomic analysis were used for normalizing phosphoproteomic and O-GlcNAc proteomic datasets. During the process of O-GlcNAc proteomics, the O-GlcNAcylation groups on proteins were biotinylated using the chemoenzymatic labeling. The O-GlcNAcylated proteins were purified using streptavidin beads followed by MS analysis. Unlabeled samples were processed in parallel, where the biotin was not attached to the O-GlcNAcylated proteins. Unlabeled samples were used for identifying the *bona fide* O-GlcNAcylated proteins during bioinformatic analysis. G in the hexagons represent the O-GlcNAc groups on proteins. Figure created using BioRender.com, licensed to the lab of J.C.C.
(TIFF)

**S3 Fig. Functional enrichment analysis of rhythmic O-GlcNAc proteome identified by label-free MS. (A–C)** Rhythmically O-GlcNAcylated proteins were entered into the Database for Annotation, Visualization and Integrated Discovery (DAVID) for enrichment of GO terms. The enriched biological processes **(A)**, molecular functions **(B)** and cellular compartments **(C)** were illustrated using bubble plots. The color of the bubbles indicates the $-\log_{10}$ false discovery rate (FDR), while the size of the bubbles indicates the number of genes in each enriched term.
(TIFF)

**S4 Fig. Examples of rhythmic O-GlcNAc proteins that regulates gene expression.** Rhythmically O-GlcNAcylated proteins (RAIN: $p < 0.05$; $q$ values were indicated) were classified based on their known association in protein complexes, and the daily rhythmicity of the oscillating subunits were plotted. Multiple protein complexes were further classified and grouped according to their biological function as indicated on the left boxes. The gray shading illustrates the dark phase of a day–night cycle. Data are presented as mean ± SEM ($n = 3$), and the blue shading area around the curve represents ±SEM.
(TIFF)

**S5 Fig. Functional enrichment analysis of O-GlcNAc proteome with day–night difference identified by TMT MS.**
**(A–C)** Rhythmically O-GlcNAcylated proteins were entered into the DAVID for enrichment of GO terms. The enriched biological processes **(A)**, molecular functions **(B)** and cellular compartments **(C)** were illustrated using bubble plots. The color of the bubbles indicates the $-\log_{10}$ false discovery rate (FDR), while the size of the bubbles indicates the number of genes in each enriched term.
(TIFF)

**S6 Fig. Functional enrichment and comparison analysis of rhythmic phosphoproteome in mouse liver nuclei.**
**(A–C)** Rhythmically phosphorylated proteins were entered into the DAVID database for enrichment of GO terms. The enriched biological processes **(A)**, molecular functions **(B)** and cellular compartments **(C)** were illustrated using bubble plots. The color of the bubbles indicates the $-\log_{10}$ false discovery rate (FDR), while the size of the bubbles indicates the number of genes in each enriched term. **(D)** Venn diagram to compare the rhythmic phosphosites newly discovered in our study versus sites previously published in two studies [56,57]. The number of rhythmic phosphosites is larger than that of rhythmic phosphopeptides, as one phosphopeptides could contain multiple phosphosites.
(TIFF)

**S7 Fig. Comparison of CLOCK detection using mass spectrometry and western blotting. (A–B)** Comparison of CLOCK O-GlcNAcylation **(A)** and protein level **(B)** detected by mass spectrometry and western blotting. The gray shading illustrates the dark phase of a day–night cycle. Data are presented as mean ± SEM, and the shading area around the curves represents SEM ($n = 2$ or 3). RAIN $p$ values for all rhythms are indicated. **(A)** $q$ value for CLOCK O-GlcNAcylation rhythm detected by global O-GlcNAc proteomics is 0.0037 (RAIN). **(B)** $q$ value for CLOCK protein rhythm detected by proteomics is 0.0039 (RAIN). The numerical values of all replicates in **(A)** and **(B)** can be found in S1 Data.
(TIFF)

**S8 Fig. Validation of CLOCK immunoprecipitation (IP) and expression. (A)** Representative blots showing that CLOCK antibody can specifically IP CLOCK protein in mouse liver. NRF ZT3 liver were used for IP with or without CLOCK antibody. **(B)** Representative blot showing that CLOCK variants expressed in HEK293T cells were at similar levels after adjusting the plasmid amount for transfection. Optimized plasmid amount was then used in experiments for Fig 5H.
(TIFF)

**S9 Fig. Metabolic pathways other than hexosamine biosynthetic pathway (HBP) that can be involved in the regulation of gene expression.** Since many metabolites were shown to regulate the function of clock proteins and other cellular proteins (Reviewed in Panda [16]), we searched for metabolites that are rhythmic under NRF condition in our metabolomic dataset, and evaluated the impact of timing of food consumption on these rhythms by comparing their rhythmicity to that under DRF. The gray shading illustrates the dark phase of a day–night cycle. Data are presented as mean ± SEM, and the shading area around the curves represents SEM ($n = 3$). $q$ values from RAIN analysis for NRF (blue) and DRF (orange) are presented under each graph.
(TIFF)

**S1 Table. Comparison between this study and published liver phosphoproteome datasets.**
(DOCX)

**S2 Table. Known functions of rhythmic phosphorylation sites that are identified in protein complexes.**
(DOCX)

**S3 Table. The domains containing or near rhythmic phosphosites with unknown function (only phosphosites identified in protein complex es are included).**
(DOCX)

**S1 File. Targeted metabolomics of NRF and DRF mouse liver for hexosamine biosynthetic pathway (HBP).** Peak intensities of metabolites detected using GC–MS and calculated concentration of HBP metabolites.
(XLSX)

**S2 File. Nuclear label-free O-GlcNAc proteome of NRF mouse liver.** Rhythmicity test of O-GlcNAc proteome.
(XLSX)

**S3 File. Nuclear proteome of NRF mouse liver.** Peptide counts and rhythmicity analysis of nuclear proteome.
(XLSX)

**S4 File. Protein complex analysis for rhythmic label-free global O-GlcNAc proteome.** List of protein complexes that contain rhythmically O-GlcNAcylated proteins, with the rhythmic subunits and the function of protein complexes indicated.
(XLSX)

**S5 File. Nuclear TMT site-specific O-GlcNAc proteome.** Normalize peptide counts and *t* test of nuclear O-GlcNAc proteome in NRF and DRF mouse liver.
(XLSX)

**S6 File. Nuclear phosphoproteome of NRF mouse liver.** (Table 1) Normalized peptide counts and rhythmicity analysis of nuclear phosphoproteome. (Table 2) Peptide-Spectrum Match (PSM) of TMT phosphoproteome for evaluating the quality of data.
(XLSX)

**S7 File. Comparison between our phosphoproteomics dataset and published rhythmic phosphoproteome.** (Table 1) Newly identified rhythmic phosphosites in our study. (Table 2) Common rhythmic phosphosites in our and published studies.
(XLSX)

**S8 File. Rhythmic kinases detected in proteomic, phosphoproteomic and O-GlcNAc proteomic datasets.** (Tables 1–3) Lists of kinases and their rhythmicity analysis from our proteomic, phosphoproteomics and O-GlcNAc proteomic datasets, respectively.
(XLSX)

**S9 File. Protein complex analysis for rhythmic phosphoproteome.** List of protein complexes that contain rhythmically phosphorylated proteins, with the rhythmic subunits and the function of protein complexes indicated.
(XLSX)

**S10 File. Analysis for G–P interaction.** (Table 1) List of proteins that showed both rhythmic O-GlcNAcylation and phosphorylation and the phase difference between these two modifications were indicated. (Table 2) List of proteins with potential G–P interplay predicted by G–P interplay motif analysis using the rhythmic phosphosites as reference. (Table 3) Predicted G–P interplay sites in close proximity (<10 amino acid) by comparing site-specific O-GlcNAc proteomics and phosphoproteomics.
(XLSX)

**S1 Data. All individual numerical values that underlie the data summarized in figures.** The numerical values of all replicates in Figs 1B, 1C (Table 1), 5D, 5F–5H (Table 2), 6C, 6G, 6H (Table 3), S7A and S7B (Table 4).
(XLSX)

**S1 Raw Images. All raw images for Western blots and gels presented in figures.**
(PDF)

## Acknowledgments

We thank the West Coast Metabolomics Center at UC Davis for their technical support.

## Author contributions

**Conceptualization:** Xianhui Liu, Yao D. Cai, Yong Zhang, Junfeng Ma, Joanna C. Chiu.

**Data curation:** Xianhui Liu, Yao D. Cai, Haiyan Zheng, Caifeng Zhao.

**Formal analysis:** Xianhui Liu, Yao D. Cai, Haiyan Zheng, Caifeng Zhao.

**Funding acquisition:** Joanna C. Chiu.

**Investigation:** Xianhui Liu, Yao D. Cai, Chunyan Hou, Xu Liu, Youcheng Luo, Aron Judd P. Mendiola, Xuehan Xu, Yige Luo, Haiyan Zheng, Caifeng Zhao, Ching-Hsuan Chen, Junfeng Ma.

**Methodology:** Xianhui Liu, Yao D. Cai, Chunyan Hou, Aron Judd P. Mendiola, Xuehan Xu, Yige Luo, Haiyan Zheng, Caifeng Zhao, Yang K. Xiang, Junfeng Ma, Joanna C. Chiu.

**Project administration:** Joanna C. Chiu.

**Supervision:** Yang K. Xiang, Junfeng Ma, Joanna C. Chiu.

**Validation:** Xianhui Liu, Yao D. Cai.

**Visualization:** Xianhui Liu, Yao D. Cai, Xu Liu, Yige Luo, Ching-Hsuan Chen.

**Writing – original draft:** Xianhui Liu, Yao D. Cai, Chunyan Hou, Junfeng Ma.

**Writing – review & editing:** Xianhui Liu, Yao D. Cai, Joanna C. Chiu.

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
