## [Editor Report · Decision Letter 0]

21 Jun 2024

Dear Dr Chiu, 

Thank you for submitting your manuscript entitled "Mealtime alters daily rhythm in nuclear O-GlcNAcome to regulate rhythmic hepatic gene expression" for consideration as a Research Article by PLOS Biology.

Your manuscript has now been evaluated by the PLOS Biology editorial staff as well as by an academic editor with relevant expertise and I am writing to let you know that we would like to send your submission out for external peer review. 

Once your full submission is complete, your paper will undergo a series of checks in preparation for peer review. After your manuscript has passed the checks it will be sent out for review. To provide the metadata for your submission, please Login to Editorial Manager (" xlink:type="simple">https://www.editorialmanager.com/pbiology) within two working days, i.e. by Jun 23 2024 11:59PM.

Kind regards,

Luke

Lucas Smith, Ph.D.

Senior Editor

PLOS Biology

lsmith@plos.org

---

## [Decision Letter · Decision Letter 1]

8 Aug 2024

Dear Dr Chiu,

Thank you for your patience while your manuscript "Mealtime alters daily rhythm in nuclear O-GlcNAcome to regulate hepatic gene expression" was peer-reviewed at PLOS Biology. Your manuscript has been evaluated by the PLOS Biology editors, an Academic Editor with relevant expertise, and by several independent reviewers.

As you will see in the reviewer reports, which can be found at the end of this email, although the reviewers find the work potentially interesting, they have also raised a substantial number of important concerns. Based on their specific comments and following discussion with the Academic Editor, it is clear that a substantial amount of work would be required to meet the criteria for publication in PLOS Biology. However, given our and the reviewer interest in your study, we would be open to inviting a comprehensive revision of the study that thoroughly addresses all the reviewers' comments. Given the extent of revision that would be needed, we cannot make a decision about publication until we have seen the revised manuscript and your response to the reviewers' comments. Your revised manuscript would need to be seen by the reviewers again, but please note that we would not engage them unless their main concerns have been addressed. 

Having discussed the reviews with the Academic Editor, we think you should address all reviewer concerns. Specifically, you should include additional analysis methods, and provide an extensive list of peptides that have been curated.

We appreciate that these requests represent a great deal of extra work, and we are willing to relax our standard revision time to allow you 6 months to revise your study. Please email us (plosbiology@plos.org) if you have any questions or concerns, or envision needing a (short) extension.

**IMPORTANT - SUBMITTING YOUR REVISION**

*Resubmission Checklist*

*Published Peer Review*

*PLOS Data Policy*

*Blot and Gel Data Policy*

Sincerely,

Suzanne

Suzanne De Bruijn, PhD, 

Associate Editor

PLOS Biology

sbruijn@plos.org

REVIEWS:

Reviewer #1: In their manuscript "Mealtime alters daily rhythm in nuclear O-GlcNAcome to regulate hepatic gene expression," Liu et al. provide evidence that nuclear protein O-GlcNAcylation in the mouse liver undergoes circadian oscillation that is disrupted by unnatural daytime feeding. Using O-GlcNAcomic and phosphoproteomic data, the authors suggest that the crosstalk between O-GlcNAcylation and phosphorylation plays a crucial role in shaping circadian hepatic gene expression and protein function. 

The nuclear characterization of the O-GlcNAcome and phosphoproteome in the liver around the clock can serve as a valuable resource, however as detailed below these analyses suffer from some major drawbacks. 

The main claim of the authors, as indicated in the title "Mealtime alters daily rhythm in nuclear O-GlcNAcome to regulate hepatic gene expression," is not supported by the data, and therefore the title is misleading. Overall, the study is primarily based on correlational data. This limitation makes it difficult to draw any definitive conclusions about the impact of circadian nuclear protein O-GlcNAcylation on hepatic gene expression as well as on other conclusions throughout the study.

Major concerns:

* There are several major issues with the statistical analyses in the study: RAIN is a very permissive analysis for detecting rhythms. It would be good to complement this analysis with JTK cycle or other relevant algorithms and the false discovery rate (FDR) should be included. The statistical analysis is flawed as the authors rely on p-values instead of q-values that correct for FDR and are widely accepted in the field (see guidelines paper PMID: 29098954). Without proper statistical analysis, the value of the study as a resource is diminished. 

* As the author state "The current technology to identify and quantify O-GlcNAc sites on proteins is not yet mature based on consultation with experts in the field, as the labile glycosidic bond of O-GlcNAcylation posed great challenge for mass spectrometry analysis," together with the batch effect issues, raise further concerns regarding the validity of the data. 

* In the section titled "Daily O-GlcNAcomics identifies rhythmically O-GlcNAcylated nuclear proteins," the authors report proteomics analyses of only 2 animals (n = 2), which corresponds to low statistical strengthes (see guidelines paper PMID: 29098954).

As listed below there are several overstatements throughout the study that are not supported by data:

* In the section titled "Crosstalk between O-GlcNAcylation and phosphorylation (G-P) regulates hepatic protein function," the authors utilize correlational evidence. Without directly testing the functional interactions through loss-of-function experiments, the correlation shown here does not definitively establish the "crosstalk" or the regulation of hepatic protein function. 

* The section titled "Daily G-P crosstalk on CLOCK and RBL2" as well contains only correlative evidence. The graph of RBL2 (Fig 5H) looks particularly unconvincing as evidence of G-P crosstalk in RBL2 protein regulation. The authors, again, provide no functional tests that directly assess these particular O-GlcNAcylations/phosphorylations. Therefore, all the conclusions drawn are merely speculative. Additional gain- or loss-of-function experiments are necessary to conclusively demonstrate causal relationships.

* In the section titled "O-GlcNAcylation of nuclear proteins in liver regulates daily hepatic transcriptome," the authors claim to assess gene expression without indicating the use of standard gene expression measurement techniques such as RNA-seq. Instead, the conclusions primarily rely on correlations between O-GlcNAcylation and phosphorylation patterns. 

* The authors "report that unnatural day time-restricted feeding (DRF) dampens O- GlcNAcylation rhythm, suggesting the disruption in liver transcriptomic rhythm previously observed in DRF condition could be mediated by dysregulation of daily protein O-GlcNAcylation rhythm." However, since DRF inverts the rhythmic transcriptome in the liver (e.g., PMID: 34059820), and dampening of the O-GlcNAcylation rhythm does not correspond to this 12 h shift, it actually argues against the authors' claim. This point merits discussion.

* The section titled "Food intake at unnatural day feeding time in mice disrupts protein O-GlcNAcylation rhythm" provides limited data and falls short of linking the observed changes in O-GlcNAcylation to specific impacts on gene expression. As Fig 7D shows the daily pattern of UDP-GlcNAc remains rhythmic under both feeding conditions, it is unclear how the authors conclude that altered UDP-GlcNAc can possibly contribute to the losss of rhytmic O-GlcNAcylation in DRF mice liver. To demonstrate causal relationships, the authors should consider a direct manipulation of UDP-GlcNAc concentration in vivo e.g., by utilizing a GFAT inhibitor.

Minor concerns:

* In the section titled "Hepatic nuclear protein O-GlcNAcylation exhibit robust daily oscillation," the authors claim that "the timing of food intake occurred earlier than the rise of UDP-GlcNAc level (feeding: ZT12-24; UDP-GlcNAc: ZT 3-7) (Fig 1C)." However, with a broad feeding time window of 12 hours, it is difficult to determine which event precedes the other. The claim should be revised.

* The authors write at the end of the first section in the results: "Our data suggest that feeding-fasting cycles could drive the oscillation of UDP-GlcNAc, which results in the oscillation of protein O- GlcNAcylation". As no manipulation of feeding was done at this stage (mice were under natural night-time feeding), the conclusion is limited to rhythmicity, and whether this is feeding-determined remains unclear at this stage of the manuscript. The text should be revised.

* Figure 3D, the authors can improve the panels. "The gray lines indicate the oscillation of each phosphopeptide, while the orange lines indicate the average phosphorylation pattern of each protein subunit." However, there is more than one protein in a panel, and it is impossible to attribute the gray lines to one or the other protein. Additionally, for TFIID, there are no gray traces? 

* The text refers to Fig 4I, which does not seem to exist. 

* The authors report on CLOCK, but what about BMAL1" BMAl1 was previously reported to be modified as well (Reference 32 in the manuscript shows a more pronounced rhythm in BMAL1 modification compared to CLOCK). 

* The IP blot in Fig 5C can be improved. It lacks a control for non-specific binding (e.g., empty beads or IgG). In its current form, the specificity of the pulled-down protein CLOCK is questionable.

Reviewer #2: In this work, Liu et al. explored the multi-omics landscape of mouse livers under night time-restricted feeding and provided a remarkable insight on the crosstalk between O-GlcNAcylation and phosphorylation in the context of meal timing. They generated a valuable resource and raised a plethora of interesting hypotheses for future research. In particular, they nailed down a new interaction between a phospho-degron and O-GlcNAcylation on CLOCK, an essential circadian clock protein. The work is technically vigorous and logically sound.

Major comments:

(1) The conclusion that "DRF dampens O-GlcNAcylation rhythm, suggesting the disruption in liver transcriptomic rhythm previously observed in DRF condition" requires additional evidence and sufficient control of the assays. Firstly, it is generally agreed that DRF reverses the liver clock as well as the hepatic rhythmic transcriptome (Refs from the labs of Schibler, Sassone-Corsi, Panda, Asher, Lazar etc.). The statement on transcriptome needs to be balanced. Secondly, Fig. 7B, Western blots of chemically labeled PTM proteins are not representative of the nuclear O-GlcNAcome. While the result implies the loss of rhythms among the most abundant nuclear PTM proteins, it does not rule out the effect of food entrainment on less abundant substrates such as CLOCK, BMAL1 and PER2. Immunoprecipitation of these transcription factors should benchmark Fig. 7B. Thirdly, in Fig. 7B-D, are NRF data shown in Fig. 1? Fourthly, UDP-GlcNAc are compartmentalized in cells. Tissue UDP-GlcNAc is not representative of nuclear UDP-GlcNAc. This is a factor that may confound some of the interpretation, i.e. O-GlcNAcome is nuclear while UDP-GlcNAc is from bulk tissue. Lastly, metabolomics may include validated metabolites with circadian rhythms under NRF and DRF. For example, orinithine, SAH etc. Nevertheless, Fig. 7 does not affect the main conclusions of this study. It can be removed without altering the flow of this work.

Minor comments:

(1) Avoid natural and unnatural when describing TRFs. Muslims practice DRF for thousands of years.

(2) Supplemental files are not accessible at this point. OGT and HCFC1 are typical O-GlcNAcylated proteins. In Fig. 2, omics profiles of OGT or HCFC1 can be shown as a quality control.

(3) TMT-phosphoproteomics has the advantage of deeper coverage, compared to label-free mass spectrometry. Fig. 3 would be strengthened by including a graph showing the number and some examples of newly identified rhythmic phosphorylation sites.

(4) In Fig. 5E-F, CLOCK Ser427 is phosphorylated by GSK3beta, not the alpha form (Cell Cycle 8:24, 4138-4146)

Reviewer #3: In the manuscript # PBIOLOGY-D-24-01760R1, Liu et al. tried to explore the correlation between daily rhythms of O-GlcNAcylation and G-P interactive phosphorylation on transcription factors and gene expression rhythms of their targets in the liver. In brief, they observed daily oscillation of global nuclear protein O-GlcNAcylation in the liver of mice subjected to natural nighttime-restricted feeding (NRF). To infer the direct and indirect roles of rhythmic O-GlcNAc changes in regulating hepatic transcriptome, they conducted multi-omics analysis, including proteomics, O-GlcNAc proteomics, and phosphoproteomics. Their O-GlcNAc proteomic analysis suggest that 11.54% of 719 O-GlcNAcylated proteins are rhythmically O-GlcNAcylated over a 24-hour day/night cycle. To investigate proteome-wide G-P crosstalk, they performed parallel TMT phosphoproteomic analysis using the same liver samples. They then proposed that rhythmic O-GlcNAcylation might indirectly modulate the hepatic transcriptome by interacting with phosphorylation. 

Overall it is an interesting study, as it suggests that mealtime may alter daily rhythm in nuclear protein O-GlcNAcylation to regulate hepatic gene expression. However, it suffers from shortcomings especially the omics data which have substantially compromised the novelty of this work and soundness of their conclusions. 

Key points: 

1. The authors performed O-GlcNAc chemoenzymatic labeling in combination with mass spectrometry proteomics to identify O-GlcNAcylated nuclear proteins in liver. However, no information on O-GlcNAc peptides is available, regardless of O-GlcNAc sites. According to this reviewer, O-GlcNAc chemoenzymatic labeling coupled with mass spectrometry shall be able to provide O-GlcNAc peptide/site information (as can be seen from several papers: PMID: 28657654, PMID: 35254053, PMID: 34161081). However, such information is not provided. After a rough look at the list of O-GlcNAc proteins identified (S2 File), it is hard to believe that some proteins are O-GlcNAcyated, for example, several immunoglobulin proteins (e.g., A0A075B5P6 and A0A0A6YX66). It will be helpful if the authors can provide at least the glycopeptide information to show that such proteins are indeed O-GlcNAc modified. Moreover, none of the proteins shown in their list (S2 File) were validated experimentally. But in Figures 4-5, it appears that the authors tried to evaluate the interaction between the O-GlcNacylation and phosphorylation on modification sites of some proteins over a day/night cycle. Clearly, O-GlcNac sites on proteins are required to do such analysis. 

2. In page 9: "Our analysis was aimed at identifying O-GlcNAcylated proteins and quantifying global O-GlcNAcylation level over the day-night cycle, rather than identifying and quantifying O-GlcNAcylation at specific residues. The current technology to identify and quantify O-GlcNAc sites on proteins is not yet mature based on consultation with experts in the field, as the labile glycosidic bond of O-GlcNAcylation posed great challenge for mass spectrometry analysis [63]". According to this reviewer, at least several quantitative O-GlcNAc proteomics studies have been reported so far (e.g., PMID: 28657654, PMID: 35254053, PMID: 34161081). Thus, it seems to be possible to do such studies. Without sites, it might be problematic to draw solid conclusions between proteome-wide G-P crosstalk. Fig. 5C provides representative blots showing the daily rhythms of CLOCK O-GlcNAcylation. However, it is plausible that there is cross-talk at the modification sites on CLOCK and RBL2 (shown in other figures).

3. The authors provided the information of phosphorylated peptides/sites (S5 File). However, key information (e.g., the localization score for the modification sites and m/z of peptides) is missing. Thus, it is hard to judge the phosphoproteomics data quality. Assuming all sites provided are of high confidence, it appears the CLOCK protein has many phosphorylation peptides/sites, but these peptides appear to have different pvalues. In addition, some peptides have two phosphorylation sites (e.g., the peptide 'ETTAQSDASEIR' shows two sites S6 and S9). Maybe the two sites have similar changes? Site-specific phosphoproteomics is very mature nowadays (e.g., PMID: 34468274, PMID: 34857927). Quantitative phosphoproteomics at the site level will clearly help improve this study. This is especially important given that bioinformatic analysis throughout the manuscript and the conclusions (e.g., G-P cross-talk) are almost exclusively based on the omics data obtained. 

Minor points: 

1. It is a bit strange the authors conducted only three biological repeats and observed batch effects on the relatively small sample sizes. Not sure whether there was big technical variability during sample processing and/or analysis. As we know, nowadays TMT can be readily used for the analysis of many samples with limited batch effects (e.g., https://pubmed.ncbi.nlm.nih.gov/32649874/) 

2. The term 'O-GlcNAcome' is awkward, which may be better described as 'O-GlcNAc proteome'.

---

## [Decision Letter · Decision Letter 2]

8 Jul 2025

Dear Dr Chiu,

Thank you for your patience while we considered your revised manuscript "Mealtime alters daily rhythm in nuclear O-GlcNAc proteome to regulate hepatic gene expression" for publication as a Research Article at PLOS Biology. Your revised study has been evaluated by the PLOS Biology editors, the Academic Editor and by two of the original reviewers. Please note, that the original reviewer 3 was not able to help assess this revision. Given that s/he had expertise in mass spec and PTM analyses and raised some important concerns in the last round of review, we opted to enlist the help of a new reviewer, with similar expertise, to help assess the revision. The new reviewer is listed as 'reviewer 4'. 

The reviews are appended below. As you will see, all three reviewers think the study has been strengthened in the revision agree that the paper offers a substantial resource for the field. However, reviewers 1 and 4 each have a number suggestions to strengthen the study further, which we think should be addressed before publication. We would like to emphasize the need to address all of Reviewer 1's lingering concerns, including with new analyses. After discussion with the academic editor, we think that the additional validation experiments suggested by Reviewer 4 would also enhance the rigor of this study and its value for the field, and so we would strongly encourage you to provide those data, if possible (although we apologize for adding some new requests at this stage in the review process). 

In light of these reviews we would like to invite you to revise the work to thoroughly address the reviewers' reports.

Given the extent of revision needed, we cannot make a decision about publication until we have seen the revised manuscript and your response to the reviewers' comments. Your revised manuscript may be sent for further evaluation by all or a subset of the reviewers.

As we think that this revision will require additional analyses and the generation of new data, we are providing a 3 month deadline for your revision. Please email us (plosbiology@plos.org) if you have any questions or concerns, or would like to request an extension. 

**IMPORTANT - SUBMITTING YOUR REVISION**

*Re-submission Checklist*

*Published Peer Review*

*PLOS Data Policy*

*Blot and Gel Data Policy*

Sincerely,

Luke

Lucas Smith, Ph.D.

Senior Editor

PLOS Biology

lsmith@plos.org

REVIEWS:

Reviewer #1: The manuscript now contains substantial improvement to the original version and may serve as a valuable resource of O-GlcNAc proteome. However, throughout the manuscript, causal language is used for phenomena that are, at present, supported by correlational evidence (see below). 

* The title claiming O-GlcNAcylation "regulate hepatic gene expression," which imply causality, should be rephrased. Furthermore, given that no transcriptomic analysis was done, the mention of "hepatic transcriptomic rhythm" in the abstract and the discussion should be removed.

* RAIN q value is inconsistently reported (some still report unadjusted p values), please provide it in all places where it is required. Given q<0.2 is quite permissive, the reader would benefit from clear indication of rhythms that has q<0.05.

* The section title "Crosstalk between O-GlcNAcylation and phosphorylation (G-P) regulates hepatic protein function" should be rephrased, as it still contains speculative correlational data and causal claims such as line 308 "inhibit each other" should be replaced with correlational claims.

* The Per2-luc assay indicates that OGT-dependent suppression of CLOCK transcriptional activity requires both S431 and S427. This is compatible with, but does not directly demonstrate, a model in which O-GlcNAc at S431 interferes with phosphorylation-dependent activation involving S427. In order to demonstrate O-GlcNAc at S431 inhibits phosphorylation, pS431 should be directly quantified in the presence/absence of OGT, using a specific phospho-antibody or mass spectrometry. Without it, the conclusion that it "demonstrate the crosstalk" or "modulates CLOCK transcriptional activity" is an overstatement, and should be rephrased.

* Line 441-442, "partially by influencing" implies causality, it should be replaced with e.g., "the change is associated with".

* Figure 4 requires more statistical tests. Spearman correlation needs to be controlled for multiple tests. The claim "in phase" "out of phase" also require appropriate test (for example, 4E is not visually apparent that the two lines are in phase). The statistical test that was used to demonstrate the day-night differences of figure 4I-L should be indicated. 

* Figure 5C is it nuclear extract or total lysate? Should be indicated, as nuclear extract should be used. 5C also lacks a loading control.

Reviewer #2: The authors have addressed all the concerns.

Reviewer #4: The paper by Liu demonstrates a periodic O-GlcNAcylation status in liver corresponding to day and night feeding. The paper has a substantial multi-Omics imprint with provides a greater depth of understanding for this pleotropic post-translational modification.

1A: The authors should perform a western blot for a nuclear protein (lamins, histones, nuclear pore proteins) to demonstrate the load control. Moreover, the authors should show the cytoplasmic fraction to verify enrichment. Does the cytoplasmic fraction show time dependent changes in O-GlcNAc as well? A simple Rl2 (anti-O-GlcNAc) western blot would show this. The argument for focusing on only the nuclear fraction does not consider transcription factors moving into and out of the nucleus. Does OGT and OGA show periodic changes in nuclear or cytoplasmic localization?

Figure 2: Its' unclear what is captured in the quantitation of the O-GlcNAc sites on peptides during the time points. Is it the O-GlcNAc is clock dependent or that the enrichment failed to capture the O-GlcNAcylated peptides. What would increase confidence is to orthogonally verify a protein that is changing during the light dark cycle. For example, EMSY O-GlcNAcylation has been validated before (Wang, Science Signaling, 2010). Ip EMSY and look for the change during the light dark cycle. That would increase rigor.

Figure 3: As above, rigor would be increased by performing a phosphorylation blot on one of the periodic phosphorylation sites.

Figure 6: As in Figure 2, showing the altered O-GlcNAcylation status via an Ip on an identified protein such as NCOR or GATAD2B would improve rigor.

Figure 6: Expression and localization data of OGT and OGA would help to identify if these proteins are different with the altered feeding schedule. The proteomics might even have the expression data?

In section headed: Daily G-P crosstalk on CLOCK, line 358 HEK cells are called "mouse HEK cells"

The additional MS methods and data interpretation are good.

---

## [Editor Report · Decision Letter 3]

27 Aug 2025

Dear Dr Chiu,

Thank you for your patience while we considered your revised manuscript "Mealtime alters daily rhythm in O-GlcNAcylation to regulate nuclear proteins involved in hepatic gene expression" for publication as a Research Article at PLOS Biology. This revised version of your manuscript has been evaluated by the PLOS Biology editors and the Academic Editor who is satisfied by the changes made in response to the last round of review. 

Based on our Academic Editor's assessment of your revision, we are likely to accept this manuscript for publication. However, before we can do so, we need you to address a few last editorial points in a revision that we anticipate will not take very long. These are detailed below. 

**IMPORTANT: Please address the following editorial requests: 

1) TITLE: We would like to suggest a change to the title which we think will more closely reflect the findings of the study. If you agree, we suggest you change it to:

"O-GlcNAcylation of nuclear proteins in the mouse liver exhibit daily oscillations that are influenced by meal timing."

(we are happy to discuss this further, if helpful)

2) DATA AVAILABILITY: Thank you for providing the raw metabolomics and proteomics datasets on Metabolomics Workbench and MassIVE. I see that the MassIVE dataset is currently set to private. Please note that this will need to be made public before publication. Also, I did not see a reviewer access token (apologies if I missed it somewhere). Can you provide me one so I can ensure that the data meets our requirements?

3) DATA AVAILABILITY: Please note that the PLOS Data Policy requires that all data be made available without restriction: http://journals.plos.org/plosbiology/s/data-availability. For more information, please also see this editorial: http://dx.doi.org/10.1371/journal.pbio.1001797

The metabolomics and proteomics data depositions will fulfill part of this requirement, but for other data that were not included in those studies (ex western blot quantifications), we ask that you provide an additional supplemental file with the underlying data. Note that we do not require all raw data. Rather, we ask that all individual quantitative observations that underlie the data summarized in the figures and results of your paper be made available in one of the following forms:

a. Supplementary files (e.g., excel). Please ensure that all data files are uploaded as 'Supporting Information' and are invariably referred to (in the manuscript, figure legends, and the Description field when uploading your files) using the following format verbatim: S1 Data, S2 Data, etc. Multiple panels of a single or even several figures can be included as multiple sheets in one excel file that is saved using exactly the following convention: S1_Data.xlsx (using an underscore).

b. Deposition in a publicly available repository. Please also provide the accession code or a reviewer link so that we may view your data before publication. 

>>Regardless of the method selected, please ensure that you provide the individual numerical values that underlie the summary data displayed in the following figure panels as they are essential for readers to assess your analysis and to reproduce it:

Fig 1B-C; Fig 5F-H; Fig 6C,G-H; Fig S6A-B

>>Please also ensure that figure legends in your manuscript include information on where the underlying data can be found, and ensure your supplemental data file/s has a legend.

>>Please ensure that your Data Statement in the submission system accurately describes where your data can be found.

4) BLOT AND GEL REPORTING REQUIREMENTS: We require the original, uncropped and minimally adjusted images supporting all blot and gel results reported in an article's figures or Supporting Information files. We will require these files before a manuscript can be accepted so please prepare and upload them now. Please carefully read our guidelines for how to prepare and upload this data: https://journals.plos.org/plosbiology/s/figures#loc-blot-and-gel-reporting-requirements

5) CODE: Per journal policy, if you have generated any custom code during the course of this investigation, please make it available without restrictions. Please ensure that the code is sufficiently well documented and reusable, and that your Data Statement in the Editorial Manager submission system accurately describes where your code can be found. 

We expect to receive your revised manuscript within two weeks. 

*Published Peer Review History*

*Press*

Sincerely,

Luke

Lucas Smith, Ph.D.

Senior Editor

lsmith@plos.org

PLOS Biology

---

## [Editor Report · Decision Letter 4]

4 Sep 2025

Dear Dr Chiu,

Thank you for the submission of your revised Research Article "O-GlcNAcylation of nuclear proteins in the mouse liver exhibit daily oscillations that are influenced by meal timing" for publication in PLOS Biology and thank you for addressing the last editorial requests in this revision. On behalf of my colleagues and the Academic Editor, Martha Merrow, I am pleased to say that we can in principle accept your manuscript for publication, provided you address any remaining formatting and reporting issues. These will be detailed in an email you should receive within 2-3 business days from our colleagues in the journal operations team; no action is required from you until then. Please note that we will not be able to formally accept your manuscript and schedule it for publication until you have completed any requested changes.

PRESS

Sincerely, 

Luke

Lucas Smith, Ph.D.

Senior Editor

PLOS Biology

lsmith@plos.org